# DiffusionPID: Interpreting Diffusion via Partial Information Decomposition

**Shaurya Dewan\***      **Rushikesh Zawar\***      **Prakanshul Saxena\***

**Yingshan Chang**      **Andrew Luo**      **Yonatan Bisk**

**Carnegie Mellon University**
`(srdewan, rzawar, prakanss, yingshac, afluo, ybisk)@andrew.cmu.edu`
(\* denotes equal contribution)

## Abstract

Text-to-image diffusion models have made significant progress in generating naturalistic images from textual inputs, and demonstrate the capacity to learn and represent complex visual-semantic relationships. While these diffusion models have achieved remarkable success, the underlying mechanisms driving their performance are not yet fully accounted for, with many unanswered questions surrounding what they learn, how they represent visual-semantic relationships, and why they sometimes fail to generalize. Our work presents Diffusion **P**artial **I**nformation **D**ecomposition (DiffusionPID), a novel technique that applies information-theoretic principles to decompose the input text prompt into its elementary components, enabling a detailed examination of how individual tokens and their interactions shape the generated image. We introduce a formal approach to analyze the uniqueness, redundancy, and synergy terms by applying PID to the denoising model at both the image and pixel level. This approach enables us to characterize how individual tokens and their interactions affect the model output. We first present a fine-grained analysis of characteristics utilized by the model to uniquely localize specific concepts, we then apply our approach in bias analysis and show it can recover gender and ethnicity biases. Finally, we use our method to visually characterize word ambiguity and similarity from the model's perspective and illustrate the efficacy of our method for prompt intervention. Our results show that PID is a potent tool for evaluating and diagnosing text-to-image diffusion models. Link to project page: https://rbz-99.github.io/Diffusion-PID/.

## 1 Introduction

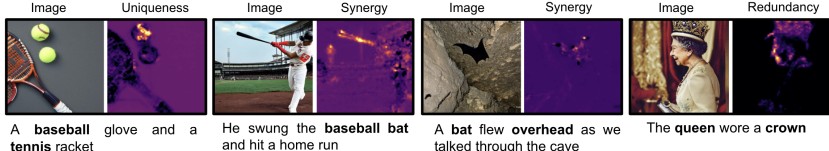

Figure 1: **Concept Figure. Left:** Our "baseball" uniqueness map specifically highlights the seam region of the tennis ball as it is visually very similar to that of a baseball, **Center:** We see a high synergy for "bat" with "baseball" and "overhead" respectively which shows that it uses these contextual cues to generate the images in the right settings, **Right:** Our redundancy map between "queen" and "crown" correctly focuses on the crown and facial region.

38th Conference on Neural Information Processing Systems (NeurIPS 2024).

The role of individual inputs and the pretraining data in diffusion-based image generation remains opaque, with little insight into how they translate often ambiguous text-based prompts into images. If we ask you to imagine a lush forest at the edge of a sandy beach, you likely fill in missing details, picturing golden sand, splashing waves, and sunlight filtering through dense green foliage. Our prior experience allows us to easily visualize complex semantic associations and imagine novel scenes.

In the previous example, you are able to both generate and justify your choices for how to complete the scene. Recent advances in diffusion-based generative models have achieved remarkable success in high-quality text-conditioned image generation, mimicking this human ability to a certain extent. However, the lack of transparency in these models raises important questions about their interpretability and controllability, making it challenging to identify biases, errors, or inconsistencies in the generated images. Text-to-image diffusion models can be brittle and prone to failure when faced with out-of-distribution inputs, ambiguous text descriptions, or nuanced contextual dependencies. This results in unrealistic, inconsistent, or even misleading generations, leading to unnatural (and linguistically non-sensical) prompt engineering [1, 2] to achieve desired results by identifying and exploiting spurious correlations captured by the model [3]. The lack of interpretability also hinders the ability to correct or refine the models when they produce undesirable outputs, limiting the potential applications of these generative models in real-world scenarios. Frequently, in multi-concept sentence-based conditioning of generative image models, concepts that are redundant to humans may be critical to the model. For instance, in the phrase "he swung a baseball bat", the word "baseball" may be redundant to humans, but it could be a crucial factor in the model's decision to generate a baseball bat instead of the animal as can be seen in Fig 1. This work posits that a crucial step towards more interpretable and robust models is to develop tools that can help explain higher-order contributions and interactions of individual words to the final image output. This would enable users to understand how the model is using the input text to generate the image, identify potential biases or errors, and refine the model's performance.

To that end, we introduce DiffusionPID, a novel approach that leverages partial information decomposition (**PID**), from information theory [4], to interpret the diffusion process and uncover the contributions of individual words and their interactions to the generated image. Compared to prior work which typically examined the contribution of tokens individually, our approach builds upon mutual information (**MI**) to examine the redundancy (**R**), synergy (**S**), and uniqueness (**U**) of the conditioning tokens. Specifically, our approach enables us to capture the interactions between concepts, revealing how individual words and phrases combine to influence the generation of specific image features. We further introduce an extension of PID conditioned on the rest of the prompt as context, Conditional PID (CPID). By quantifying these interactions, DiffusionPID provides a more comprehensive understanding of the diffusion process, enabling the development of more interpretable and controllable text-to-image models.

This work contributes: **(1)** a method to compute the image and pixel-level PID of diffusion-generated images with respect to concepts from the input prompt; **(2)** an extension of PID, namely CPID, to better account for the context provided by the rest of the prompt during the generative process; **(3)** an evaluation of the DiffusionPID framework for elucidating model dynamics (via **R, S, U**) on complex linguistic concepts such as synonyms and homonyms and exposing ethnic/gender bias in the model; and finally, **(4)** a method of prompt intervention via PID to remove redundant words and retain unique and synergistic words.

## 2   Related Work

**Text-to-Image (T2I) Diffusion Models.**   Diffusion-based text-to-image models have shown impressive results in image synthesis [5, 6, 7, 8]. These models are capable of generating images that are both realistic and semantically consistent with the input text. This has led to the development of many large-scale and diverse datasets [9, 10, 11]. Recent datasets of challenging prompts and benchmarks based on visual question answering [12] and scene graphs have enabled researchers to assess and refine their models [13, 14, 15, 16]. Inspired by the observation that existing models struggle to accurately bind attributes to objects [17], some approaches propose to modify the cross-attention layers by explicitly amplifying the token-wise attention used for text conditioning [18, 19, 20, 21, 22], or by composing multiple outputs to generate semantically compositional images [23]. These methods aim to improve the model's ability to capture fine-grained semantic relationships between text and image. These diverse approaches demonstrate the ongoing effort to develop more effective and flexible text-to-image synthesis models and highlight the need for continued research in this area.

**Interpretability in Deep Learning:** Interpretability is key to understanding and trusting the decisions made by complex models. LIME [24] presents a technique for interpreting individual predictions of any machine learning (ML) model via local linear models, while SHAP [25] assigns each feature an importance value based on Shapley values from cooperative game theory. CAM [26] and Grad-CAM [27] localize image regions in a CNN using a weighted sum of intermediate layers' activations. Network Dissection [28] operates at the neuron level, tracking image regions corresponding to an excited neuron across all the dataset images, which GAN Dissection [29] extends for GANs. MILAN [30] and CLIP-Dissect [31] identify semantic concepts for an individual neuron. Benchmarks such as ARO [32], Winoground [33], EQBEN [34], and WHOOPS [35] introduce datasets to specifically probe the compositional and semantic understanding of and show that most SOTA models struggle with relational concepts. Some works have also explored the application of PID in different settings [36]. [37] uses PID to measure the contribution of each neuron towards a target concept and [38] applies PID on generic multimodal datasets and models. Finally, MULTIVIZ [39] analyzes the behavior of multimodal models, specifically focusing on the unimodal, cross-modal, and multi-modal contributions. Overall, a lot of progress has been made in understanding and interpreting ML models. Given the recent explosion of work on diffusion models, we primarily focus on interpreting these models.

**Interpretability of Diffusion Models:** Interpretability of diffusion models remains an open challenge. Data attribution [40, 41, 42, 43] is a commonly used approach to identify training samples most responsible for the appearance of a generated image. Similarly, [44, 45] decompose the images associated with a particular concept into the implicit semantic concepts the generative model considers to be most similar. Others [46, 47, 48, 49, 50] have discovered a semantic latent space for diffusion models such that modifications in this space edit attributes of the original generated images. [51, 52] are similar but operate in the text embedding space. Recently, there have also been works that try to gain a neuron-level and network-level understanding of these models. For instance, [53] divides the latent vector into multiple groups of elements and designs different noise schedules for each group so that each group controls only certain elements of data, explicitly giving interpretable meaning. [54] identifies clusters of neurons corresponding to a given concept with gradient statistics. [52, 55] show that diffusion models struggle on prompts containing homonyms (words with multiple meanings) and [56, 57] suggested the presence of ethnic, gender, and semantic biases in diffusion models – we investigate both of these conditions in sections 4.3 and [4.1, 4.2], respectively. Recently, DAAM [58] proposed to analyze the cross-attention layers of diffusion models to produce attribution maps for tokens from the input prompt. However, these maps are generally very noisy and offer limited information if the model fails to generate a given concept. [59] introduced a method to compute the mutual information (MI) and conditional mutual information (CMI) between the input text prompt and each generated pixel. While their work provides stronger and more meaningful insights, their approach only provides coarse information on the model. Our work enables fine-grained model understanding by decomposing the mutual information between the text prompt and image output by using partial information decomposition (PID).

## 3 Method

Image diffusion models learn to model an image distribution, $X$, through a progressive noise addition and removal process. The forward diffusion process, defined by $x_\alpha = \sqrt{\sigma(\alpha)}x + \sqrt{\sigma(-\alpha)}\epsilon$, gradually adds Gaussian noise, $\epsilon \sim N(0, I)$, to the image, $x \sim p(X)$, over a sequence of timesteps, where $\alpha$ denotes the log SNR/noise schedule and $\sigma$ is the sigmoid function. A model learns to reverse this process, denoising the data to gradually transform the noise back into samples from the original data distribution. This denoising process has been applied to text-to-image generation with incredible success by conditioning the process on text prompts $y$. We employ the denoising process to measure the influence of individual tokens (from the input prompt) and their interactions on the resulting generation. We consider the diffusion model to be an optimal denoiser that can predict the added ground-truth noise, $\epsilon$, at noise level $\alpha$ as:

$$\hat{\epsilon}_\alpha(x) = \underset{\hat{\epsilon}(.)}{\arg\min} \, E_{p(x),p(\epsilon)}[\|\epsilon - \hat{\epsilon}_\alpha(x_\alpha)\|^2] \tag{1}$$

We follow [59]'s definition of mutual information (MI) and conditional mutual information (CMI) computed using a diffusion model. Extending Eq 1 to the conditional case where we assume $\hat{\epsilon}_\alpha(x_\alpha|y)$ is the optimal denoiser for the conditional distribution $p(x|y)$, the Log Likelihood Ratio (LLR), which is also the mutual information (MI) $i(y; x)$, between a pixel $x$ and an input text prompt $y$ is defined as:

$$\log p(x \mid y) - \log p(x) = \frac{1}{2} \int E_{p(\epsilon)}[||\epsilon - \hat{\epsilon}_\alpha(x_\alpha)||^2 - ||\epsilon - \hat{\epsilon}_\alpha(x_\alpha \mid y)||^2]d\alpha \qquad (2)$$

$$\log p(x \mid y) - \log p(x) = \frac{1}{2} \int E_{p(\epsilon)}[||\hat{\epsilon}_\alpha(x_\alpha) - \hat{\epsilon}_\alpha(x_\alpha \mid y)||^2]d\alpha \qquad (3)$$

Eq 3 is derived from Eq 2 using the orthogonality principle and the derivation can be found in the supplemental. We also provide graphs comparing the MMSE estimates obtained from Eq 2 versus the simplified form in Eq 3 for varying levels of noise/SNR and discuss the uncertainty in our estimator in the supplemental.

Essentially, MI quantifies the information an input variable $y$ individually provides about an output variable $x$. This definition of pixel-wise MI derived from [60] can be intuitive and easily understood as the quantity by which the text prompt $y$ increases the probability of observing $x$ relative to the prior probability without text conditioning. The same work provides an alternate definition for MI as:

$$i(y; x) = -\log p(y) + \log p(y|x) = \log p(x|y) - \log p(x) \qquad (4)$$

Similarly for the CMI of phrase $y_1$ given the context of the phrase $y_2$ in the input text prompt $y$:

$$i(y_1; x|y_2) = -\log p(y_1|y_2) + \log p(y_1|x, y_2) = \log p(x|y_1, y_2) - \log p(x|y_2) \qquad (5)$$

From Eq 5, CMI can be understood as the information that $y_1$ contains about $x$ beyond what is already contained in $y_2$. Following [60], we provide the mathematical definitions for the various terms in the partial information decomposition (PID) at the pixel-level. We start with the MI of two input events (phrases) with an output variable (pixel) defined as:

$$i(y_1, y_2; x) = r(y_1, y_2; x) + u(y_1 \backslash y_2; x) + u(y_2 \backslash y_1; x) + s(y_1, y_2; x) \qquad (6)$$

Here, $y_1 \backslash y_2$ means $y_1$ excluding $y_2$. $r(y_1, y_2; x)$ is the redundant or overlapping information between $y_1$ and $y_2$ (redundancy), $u(y_1 \backslash y_2; x)$ is the unique information contributed by $y_1$, $u(y_2 \backslash y_1; x)$ is the unique information contributed by $y_2$, and $s(y_1, y_2; x)$ is the new information contributed by the combination of $y_1$ and $y_2$ (synergy) that neither of them could contribute on their own. It is important to note that the uniqueness defined here is with respect to the other input variable, i.e., it is the unique information the given phrase contributes that the other phrase does not.

A natural measure of redundancy is the expected value of the minimum information that any source provides about each outcome of $x$. This captures the idea that redundancy is the information common to all sources (the minimum information that any source provides) while taking into account that sources may provide information about different outcomes of $x$. However, the redundant information must capture when two predictor variables are carrying the same information about the target, not merely the same amount of information. This means that the sign/direction of information matters. To account for this and to ensure the definition complies with all the required axioms on redundancy, the redundancy is broken down into a positive $r^+(y_1, y_2; x)$ and negative $r^-(y_1, y_2; x)$ component associated with the informative $i^+(y_i; x)$ and misinformative $i^-(y_i; x)$ MI terms respectively. Thus, the equation for computing the redundancy can be derived as:

$$r^+(y_1, y_2; x) = \min_{y_i} i^+(y_i; x) = \min_{y_i}[-\log p(y_i)], \quad y_i \in \{y_1, y_2\} \qquad (7)$$

$$r^-(y_1, y_2; x) = \min_{y_i} i^-(y_i; x) = \min_{y_i}[-\log p(y_i|x)] \qquad (8)$$

$$r(y_1, y_2; x) = r^+ - r^- = \min_{y_i}[-\log p(y_i)] - \min_{y_i}[-\log p(y_i|x)] \qquad (9)$$

$$r(y_1, y_2; x) = \min_{y_i}[-\log p(y_i)] - \min_{y_i}[-\log p(x|y_i) + \log p(x) - \log p(y_i)] \qquad (10)$$

To compute the probability of an input event (phrase) $p(y)$ in the above equations, we make use of BERT [61]. One of the tasks BERT was trained on was to predict a masked token by predicting the probabilities of all possible tokens from a fixed dictionary and taking the maximum. We use this to obtain the probability of a given phrase. In the unconditional case, we only require the objective probability of the phrase with no other tokens except the special [MASK] token present in the string fed to BERT.

Given the definitions for MI and redundancy, the uniqueness of an input event/phrase and the synergy between phrases can be derived as:

$$u(y_1 \backslash y_2; x) = i(y_1; x) - r(y_1, y_2; x) \qquad (11)$$

$$s(y_1, y_2; x) = i(y_1, y_2; x) - r(y_1, y_2; x) - u(y_1 \backslash y_2; x) - u(y_2 \backslash y_1; x) \qquad (12)$$

We also introduce CPID (Conditional PID) as an extension of PID to take the context contributed by the rest of the prompt into consideration. It represents the PID for the conditional case where all the PID components and probability terms are now conditioned on the rest of the prompt. This is similar to the CMI extension of MI in [59]. We rewrite equations 6, 10, 11 and 12 with the required changes below for easy reference ($y$ signifies the rest of the prompt with the terms $y_1$ and $y_2$ removed):

$$i(y_1, y_2; x|y) = r(y_1, y_2; x|y) + u(y_1 \backslash y_2; x|y) + u(y_2 \backslash y_1; x|y) + s(y_1, y_2; x|y) \qquad (13)$$

$$r(y_1, y_2; x|y) = \min_{y_i}[-\log\ p(y_i|y)] - \min_{y_i}[-\log\ p(x|y_i, y) + \log\ p(x|y) - \log\ p(y_i|y)] \qquad (14)$$

$$u(y_1 \backslash y_2; x|y) = i(y_1; x|y) - r(y_1, y_2; x|y) \qquad (15)$$

$$s(y_1, y_2; x|y) = i(y_1, y_2; x|y) - r(y_1, y_2; x|y) - u(y_1 \backslash y_2; x|y) - u(y_2 \backslash y_1; x|y) \qquad (16)$$

It is important to note that all our definitions for all the information terms require the evaluation of an integral over an infinite range of SNRs. In practice, we make use of truncated logistic-based importance sampling to evaluate this integral, similar to [59].

## 4 Experiments

We use our method to conduct a detailed analysis of diffusion models in various situations. We introduce several tasks along with corresponding datasets to study these models' behavior in depth. For our experiments, we primarily focus on the pre-trained Stable Diffusion 2.1 model from Hugging Face. Latent diffusion models encode images into a lower dimensional latent space before the denoising process. Thus, all our PID computations occur in this latent space, i.e., we consider the image in this lower dimensional space as $x$. During visualization, the heatmaps are bilinearly interpolated from this latent space to the original image resolution. Finally, we make use of 50 samples for evaluating the integral over SNRs using importance sampling. A single A6000 GPU was used to generate the PID maps for each data sample.

### 4.1 Gender Bias

**Setup**: Gender is a social construct and a complex study in its own right. In this work, we search for the perpetuation of traditional gender roles and associations. While there is a rich literature on the implications of such biases and how models can exacerbate them [62, 63, 64, 65, 66, 67, 68, 69, 70], we simply present PID as another tool in that investigative process. In this study, we evaluate whether the diffusion model exhibits gender bias in generating images of people in various occupations. Our objective is to determine if the model consistently associates specific genders with certain occupations. For this task, we take a set of 188 common occupations such as professor, valet, receptionist, janitor, etc., and create 376 prompts (188 x 2) by joining the occupation with each gender, male or female, one at a time. More details on the dataset creation process can be found in the supplemental. To check for bias, we compare the image-level redundancy values of each gender with the occupations. The image-level value is simply computed by using the expectation of the MSE term over all pixels instead of the pixel-level MSE values in our equations. The final redundancy values are normalized across the entire dataset to the range $[0, 1]$. We hypothesize that whenever the redundancy of one gender is much higher than that of the other with the occupation, the model is biased towards the higher redundancy gender for that occupation.

**Results**: We can see in Table 1 that vocations that are typically stereotyped to be performed by males such as plumber, carpenter, and police officer, have high redundancy values for the male gender as compared to the female gender. Similarly, we see higher redundancy for the female gender for female stereotyped jobs such as babysitter and teacher. We also observe that the average redundancy for females across all occupations is very low, indicating significant model bias against generating females for any occupation. Thus, it is clear that the model has learnt gender biases.

| Occupation | Male | Female |
|---|---|---|
| Surgeon | **0.539** | 0.055 |
| Soldier | **0.250** | 0.136 |
| Judge | **0.304** | 0.286 |
| Doctor | **0.871** | 0.090 |
| Plumber | **0.605** | 0.038 |
| Carpenter | **0.365** | 0.093 |
| Police Officer | **0.390** | 0.091 |
| Babysitter | 0.240 | **0.531** |
| Teacher | 0.098 | **0.419** |
| Average | 0.286 | 0.194 |

Table 1: **Gender Bias:** Redundancy between gender and occupation

| Occupation | Black | Asian | Caucasian | Hispanic |
|---|---|---|---|---|
| Athlete | **0.321** | 0.132 | 0.167 | 0.156 |
| Artist | **0.106** | 0.069 | 0.062 | 0.045 |
| Engineer | 0.126 | **0.156** | 0.080 | 0.097 |
| Physicist | 0.109 | **0.209** | 0.162 | 0.064 |
| Butcher | 0.110 | 0.179 | **0.474** | 0.396 |
| Coach | 0.118 | 0.107 | **0.433** | 0.128 |
| Nurse | 0.106 | 0.106 | 0.127 | **0.219** |
| Agriculturist | 0.046 | 0.117 | 0.337 | **0.450** |
| Average | 0.133 | 0.233 | 0.255 | 0.236 |

Table 2: **Ethnicity Bias:** Redundancy between ethnicity and occupation

## 4.2 Ethnic Bias

**Setup**: Similar to gender, race and ethnicity are not hard and fast classes, but indicate how individuals self-identify. Previous work [71, 69, 72, 73, 74, 75] has shown that deep learning models are prone to learning racial and ethnic biases, especially when the underlying training data is imbalanced. With the recent shift towards diffusion-generated synthetic datasets [9, 10, 11], it has become crucial to test for the presence of such biases in diffusion. Similar to the experiment above, we examine the diffusion model for ethnic bias to see if it associates specific ethnicities with specific occupations. We use the same 188 occupations used in the gender bias experiment but now we combine them with each ethnicity (Caucasian, Black, Asian, and Hispanic) instead of gender. We end up creating a dataset of 752 (188 x 4) image-prompt pairs. Refer to the supplemental for more details on the dataset creation process. The same approach is adopted as the previous experiment where we compare the normalized image-level redundancy values of the various ethnicities with the occupations.

**Results**: We can see from Table 2 that jobs that are commonly stereotyped for certain ethnicities such as athlete with Black, engineer with Asian, and so on, have higher redundancies with those respective ethnicities. We also compare the redundancies of each ethnicity averaged over all occupations and observe a very low value for the Black ethnic group which means that the model is heavily biased against this group and is less likely to generate people from this group for any occupation on average. Thus, we clearly see that the model has learnt biases on the basis of ethnicity.

## 4.3 Homonyms

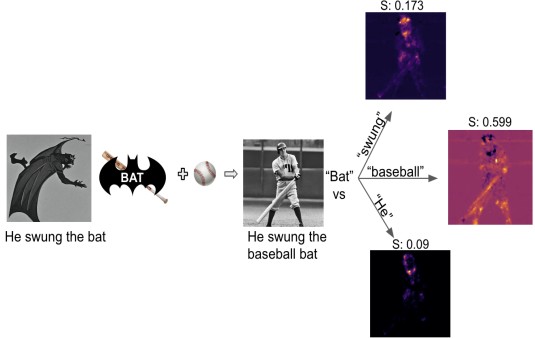

Figure 2: **Homonyms.** We see that the word "baseball" provides the required synergistic context with the homonym "bat" to pick the sports setting over the animal. This effect can be confirmed in the synergy maps and image-level synergy values (S) as well where we observe a high synergy for "bat" with "baseball" compared to other words such as "He" and "swung".

A diffusion model can behave in different manners when faced with prompts containing homonyms in different contexts. A word is a homonym when it can take on different meanings when used with

different contextual words which we term as *modifiers*. For example, the phrases "football **match**" and "he lit a **match**" should generate two completely different visuals. We examine how diffusion models handle such prompts in this experiment.

**Setup**: For this task, we use our approach to analyze the novel information contributed towards the image generation process by different *modifiers*. We create a dataset of 242 prompts containing homonyms in different contexts. We study the synergy map between the homonyms and the *modifiers* to see if the model is able to extract new information from their combination to generate an image of the homonym in the right context.

**Results**: In the left figure of Fig 3, we can see our synergy maps correctly highlight the area pertaining to the new information contributed by the combination of the homonym ("bowl") and modifier ("ceramic" and "game" respectively). However, in the right figure of Fig 3, the model fails to generate the homonym ("mole") in the right context even when given the appropriate modifiers ("coworker" and "searching"). This failure can be understood by the low synergy maps which reveal that the model fails to learn the synergistic relationship between the homonym and the modifiers. Thus, the synergy maps help us decode these kinds of failures of the model.

We also provide synergy map visuals obtained from our CPID implementation in the supplemental.

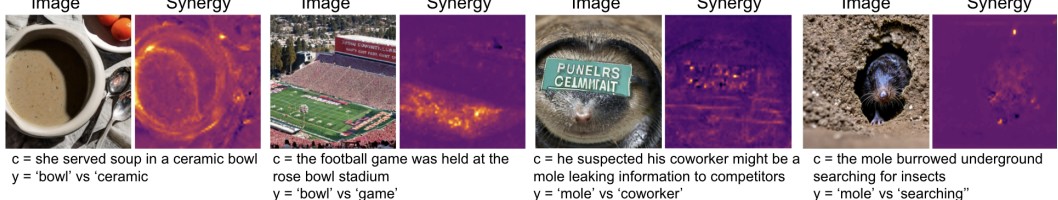

c = she served soup in a ceramic bowl
y = 'bowl' vs 'ceramic

c = the football game was held at the rose bowl stadium
y = 'bowl' vs 'game'

c = he suspected his coworker might be a mole leaking information to competitors
y = 'mole' vs 'coworker'

c = the mole burrowed underground searching for insects
y = 'mole' vs 'searching"

Figure 3: **Homonyms. Left:** Successful generation of homonym "bowl" in different contexts due to high synergy with modifiers "bowl" and "game". **Right:** Failure case where the model generates the homonym "mole" with the same semantic meaning, the animal, due to its failure to use contextual information from words like "coworker" as can be seen in the synergy map.

### 4.4 Synonyms

**Setup**: Dealing with synonyms, i.e., different words with the same meaning, can be challenging for diffusion models. Here, we probe the diffusion model with prompts containing known synonym pairs to see how it handles them and if it correctly identifies the given words as synonyms. To that end, we generated a dataset of images from 132 text prompts containing synonym pairs. More details on the dataset creation process can be found in the supplementary. Our hypothesis is that the redundancy will be high between words that the model considers as synonyms.

**Results**: In Fig 4, we can see how our approach very clearly outputs high redundancy in the region of the object that both synonyms, "cube" and "cuboid", refer to. Similarly, the redundancy map for the words "bed" and "mattress" correctly highlights the bed region. To have a fair comparison with MI, CMI, and DAAM, we take the intersection between the maps of the individual synonyms produced by each of these methods (details in supplementary). It is visible from Fig 4 that these methods fail to find the region of interest and produce very sparse maps. Hence, our method is better suited for interpreting what words/phrases the diffusion model considers semantically similar.

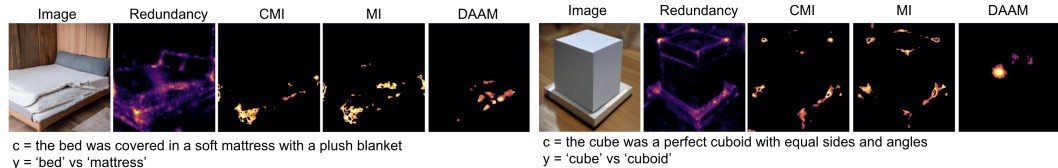

c = the bed was covered in a soft mattress with a plush blanket
y = 'bed' vs 'mattress'

c = the cube was a perfect cuboid with equal sides and angles
y = 'cube' vs 'cuboid'

Figure 4: **Synonyms.** Our redundancy map is able to highlight that the model considers the pairs "bed" and "mattress" (**left**) and "cube" and "cuboid" (**right**) as semantically similar.

### 4.5 Co-Hyponyms

Previous work has shown that diffusion models sometimes fail to generate certain objects mentioned in the prompt. One such set of cases the model struggles with is prompts containing co-hyponyms,

i.e., concepts that are semantically very similar but not identical. The model tends to generate only one of the objects rather than both of them or a fused version of the objects. We hypothesize that this phenomenon occurs because the model confuses the two concepts to carry the same semantic meaning even if they don't for us as human beings. This means that the redundancy between them is expected to be very high which is also what we can see in our experiments. We introduce 2 prompt datasets containing co-hyponyms and are discussed below.

### 4.5.1 Co-Hyponym COCO

**Setup**: To construct this dataset, we used the COCO dataset's [76] super-category hierarchy to extract co-hyponym pairs. The supplemental contains more details on the dataset creation process.

**Results**: In Fig 5, the redundancy is visibly high in the region where the diffusion model has fused features from the words "cat" and "elephant". Similarly, the redundancy is highly activated in the region of the only generated object due to the high semantic similarity of the words "pizza" and "sandwich". Thus, it is apparent that for cases where the diffusion model fails to learn the difference in the semantic meaning of two co-hyponyms, it confuses the two to be referring to the same concept, signified by the high redundancy, and fails to generate one of the objects. Similar to the synonym experiment above, we compare our redundancy map with the intersection maps from MI, CMI, and DAAM and observe that they provide little information for interpreting the reason for the model's failure. Thus, the redundancy map proves to be a useful tool to understand why the model fails here.

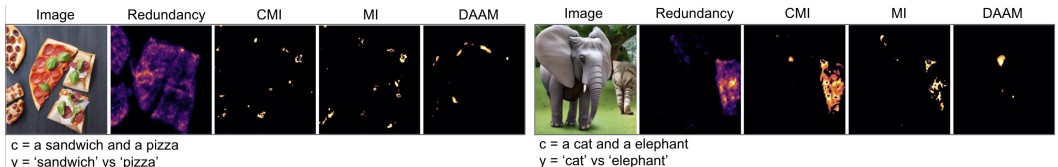

Figure 5: **COCO Co-Hyponyms.** The redundancy map proves to be very useful in finding out the reason behind the model's failures in these figures. It confuses the co-hyponym pairs ("sandwich", "pizza")(**left**) and ("elephant", "cat")(**right**) to have the same meaning for the co-hyponyms as seen from the redundancy maps, which results in erroneous generations.

### 4.5.2 Co-Hyponym WORDNET

**Setup**: We make use of Wordnet's [77] hyponym and hypernym relations between the synsets of the words to obtain 798 co-hyponym pairs for the prompts. Refer to the supplementary for more details on the dataset creation process.

**Results**: In the left figure of Fig 6, we can see the fusion of features from both objects mentioned in the prompt, while in the right figure, we see only one of the objects has been generated. The redundancy maps correctly highlight the image regions pertaining to both the co-hyponyms. Again it is observed that MI, CMI, and DAAM only very sparsely highlight the correct region of overlap. This reinforces our findings from the previous COCO-based experiment that the model mixes up the co-hyponyms and our redundancy map from PID helps pin down the reason behind this phenomenon.

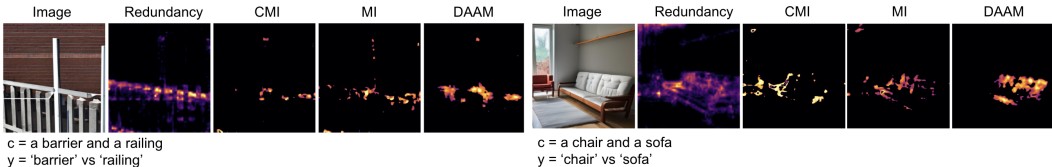

Figure 6: **Wordnet Co-Hyponyms.** Similar to the COCO Co-Hyponyms dataset, the redundancy maps tell us that the model confuses the co-hyponym pairs ("barrier", "railing")(**left**) and ("chair", "sofa")(**right**) to have the same meaning for the co-hyponyms, resulting in erroneous generations.

### 4.6 Prompt Intervention

We use our method to identify redundant words in the prompts, remove them, and verify that this intervention results in little change to the image. We test this task on all the datasets involving

redundant terms: co-hyponym wordnet, co-hyponym coco, synonyms, and occupation bias (ethnic and gender). For more details on how we edit the image, refer to the supplementary.

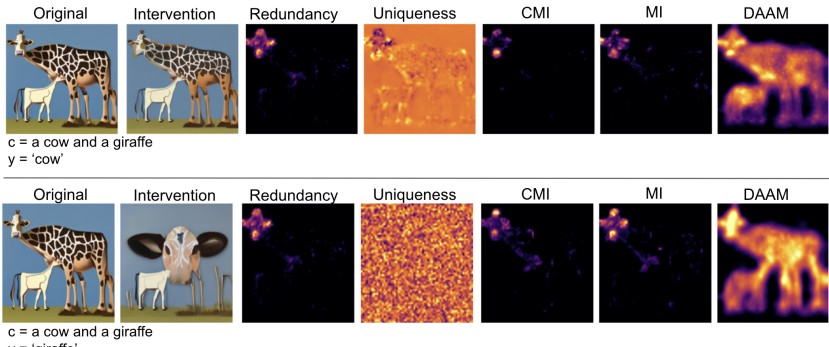

Figure 7: **Prompt Intervention.** The redundancy is highly activated in the face region of the giraffe. On a closer look, we see that the face is that of a cow meaning that the word "cow" is redundant. This is confirmed as the image changes very little on omitting it from the prompt.

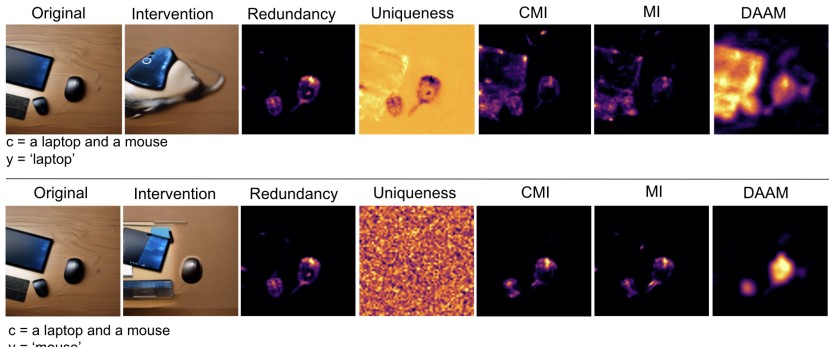

Figure 8: **Prompt Intervention.** The redundancy is highly activated in the mouse region meaning that the word "mouse" is redundant. The uniqueness of "mouse" is also very low and spread out. This redundancy is confirmed as the image changes very little on omitting it from the prompt.

Both figures 7 and 8 clearly show that whenever the redundancy is high in a region corresponding to a particular object and the uniqueness for that object's term is low/spread out, then removing that object has minimal effect on the image. On the other hand, we see that the MI, CMI, and DAAM maps are highly activated for these redundant terms, indicating that these metrics are not ideal for identifying redundant terms in the input prompt. For instance, in the first figure, the redundancy map is highly activated in the mouse region and the uniqueness of the term "mouse" is spread out. Removing the word "mouse" results in an image very similar to the original whereas removing the word "laptop" drastically changes the image. The MI, CMI, and DAAM maps are notably high for the "mouse" term, incorrectly suggesting that removing this term should have significantly altered the image. A similar effect can be observed for the term "cow" in the second figure.

### 4.7 Most Representative Features

The most representative features of a concept are those characteristics that uniquely define it from the model's perspective. Given a set of classes, the uniqueness map obtained from PID can be used to localize these features. As we can see in Fig 9, when the input prompt is "hair dryer and toothbrush", even though the image is a mix of a hair dryer and a toothbrush, the uniqueness map highlights the region corresponding to the toothbrush bristles. Similarly, we can see the bear's facial features being highlighted correctly in the other example. Thus, our uniqueness maps prove useful for this task of identifying the unique characteristics of an object.

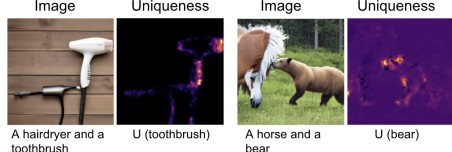

Figure 9: **Most Representative Features. Left:** The "toothbrush" uniqueness map correctly captures the toothbrush bristles, their most distinct feature. **Right:** The "bear" uniqueness map correctly captures the bear region, specifically the face.

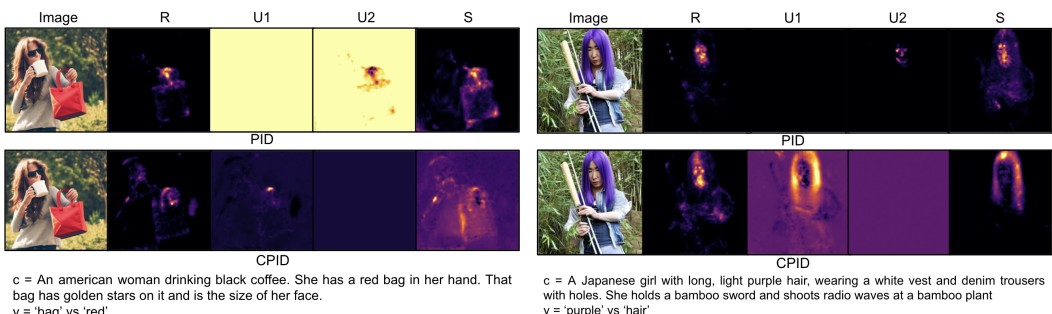

c = An american woman drinking black coffee. She has a red bag in her hand. That bag has golden stars on it and is the size of her face.
y = 'bag' vs 'red'

c = A Japanese girl with long, light purple hair, wearing a white vest and denim trousers with holes. She holds a bamboo sword and shoots radio waves at a bamboo plant
y = 'purple' vs 'hair'

Figure 10: Results of PID and CPID on complex prompts

## 4.8 Complex Prompts

In this experiment, we test the efficacy of our method on more complex prompts, similar to those in the diffusion training distribution. These prompts usually mention several objects and their corresponding attributes. We find that our method remains effective and informative even in these challenging examples. We visualize the information maps between objects and attribute-defining terms.

In both visuals in Fig 10, we observe a high synergy because the attribute modifies the object's visual properties in some form. We also see that CPID provides slightly better localized results than its PID counterpart in practice. This is expected as CPID accounts for the contextual contribution of the rest of the prompt in the image generation process and better captures the specific contribution of the two terms under consideration.

## 5 Discussion

**Limitations and Future Work:** Although we have exhibited the efficacy and benefits of Diffusion-PID, there remains some scope for improvement that future work can explore. In all our experiments, we compute the PID terms only for two phrases from the prompt but PID can be extended to more than two input variables. Another interesting research direction could be to compute PID on other, non-diffusion-based models. Also, our approach requires access to the diffusion model, making it difficult to apply it to closed-source models, which means that alternative methods of computing PID need to be explored. There are also some other applications that could be tried with PID such as using the uniqueness information to localize distinctive features to differentiate closely related classes in image classification. Recently there have been works such as [78] that integrate other forms of conditioning such as masks to enable better control of diffusion models. PID-based analysis of these multimodal forms of conditioning is another worthwhile research direction for the future.

**Conclusion:** The fine-grained breakdown of MI afforded by our approach, DiffusionPID, allows us to understand the decision-making process of diffusion models, identify their shortcomings, and pin down the reasons for their failures. This understanding is critical given the misalignment between the model's world understanding and ours. We can use the insights furnished from our approach to make diffusion models better aligned with humans' conceptual understanding and counter its limitations and biases. Our work can serve as a strong fundamental basis for further research on using information theoretic concepts such as PID to dissect, study, and enhance deep learning models.

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

# 6 Appendix

## 6.1 MI and PID

Here, we explain the concepts of mutual information and partial information decomposition in greater detail.

**Mutual Information (MI)** is a measure that estimates the amount of information obtained for an output variable from a predictor variable passed through a function. It measures to what extent an input variable reduces the uncertainty in the model's prediction. In deep learning, where understanding the dependencies and mapping between input features and target variables is vital, MI serves as a powerful tool to quantify these dependencies accurately.

**Partial Information Decomposition (PID)**, as defined by [60], shows that two predictor variables $X_1$ and $X_2$ can contribute information about a target variable $Y$ in four possible ways. $X_1$ can provide some information about $Y$ independently/uniquely, $X_2$ too can provide some information about $Y$ independently/uniquely, both $X_1$ & $X_2$ can contribute some overlapping/redundant information, i.e., some of the information contributed by both is the same, and $X_1$ & $X_2$ can contribute synergistic information which is information neither of them can contribute on their own and only in conjunction with the other. [60] also state that mutual information can be derived from first principles as fundamentally pointwise quantities where they measure the information content of individual events. The overall (average) mutual information can then be calculated by taking the expectation over all events for the relevant variables.

## 6.2 Simplification of Mutual Information

Here we outline the derivation of Eq 3 from Eq 2 using the orthogonality principle:

$$I(X;Y) = \mathbb{E}_{p(x,y)}[\log p(x|y) - \log p(x)] \tag{1}$$

$$I(X;Y) = \mathbb{E}_{p(x,y)}[\frac{1}{2}\int \mathbb{E}_{p(\epsilon)}[||\epsilon - \hat{\epsilon}_\alpha(x_\alpha)||^2 - ||\epsilon - \hat{\epsilon}_\alpha(x_\alpha|y)||^2]d\alpha] \tag{2}$$

By expanding all the squares and re-arranging we get:

$$I(X;Y) = \mathbb{E}_{p(x,y)}\left[\frac{1}{2}\int \mathbb{E}_{p(\epsilon)}\left[||\hat{\epsilon}_\alpha(x_\alpha) - \hat{\epsilon}_\alpha(x_\alpha|y)||^2\right]d\alpha\right] +$$
$$2\mathbb{E}_{p(y)}\left[\frac{1}{2}\int \mathbb{E}_{p(x|y),p(\epsilon)}\left[(\hat{\epsilon}_\alpha(x_\alpha) - \hat{\epsilon}_\alpha(x_\alpha|y))\cdot(\hat{\epsilon}_\alpha(x_\alpha|y) - \epsilon)\right]d\alpha\right] \tag{3}$$

Here,

$$+2\mathbb{E}_{p(y)}\left[\frac{1}{2}\int \mathbb{E}_{p(x|y),p(\epsilon)}\left[(\hat{\epsilon}_\alpha(x_\alpha) - \hat{\epsilon}_\alpha(x_\alpha|y))\cdot(\hat{\epsilon}_\alpha(x_\alpha|y) - \epsilon)\right]d\alpha\right] \equiv \ominus \tag{4}$$

Based on the orthogonality principle [1], which states:

$$\forall f, \quad \mathbb{E}_{p(x|y)p(\epsilon)}\left[f(x_\alpha,y)\cdot(\hat{\epsilon}_\alpha(x_\alpha|y) - \epsilon)\right] = 0 \tag{5}$$

The term $(\hat{\epsilon}_\alpha(x_\alpha|y) - \epsilon)$ represents the error of the MMSE estimator, which is orthogonal to any estimator. Therefore, the second term becomes zero, leading to:

$$I(X;Y) = \mathbb{E}_{p(x,y)}\left[\frac{1}{2}\int \mathbb{E}_{p(\epsilon)}\left[||\hat{\epsilon}_\alpha(x_\alpha) - \hat{\epsilon}_\alpha(x_\alpha|y)||^2\right]d\alpha\right] \tag{6}$$

This is the same as Eq 3.

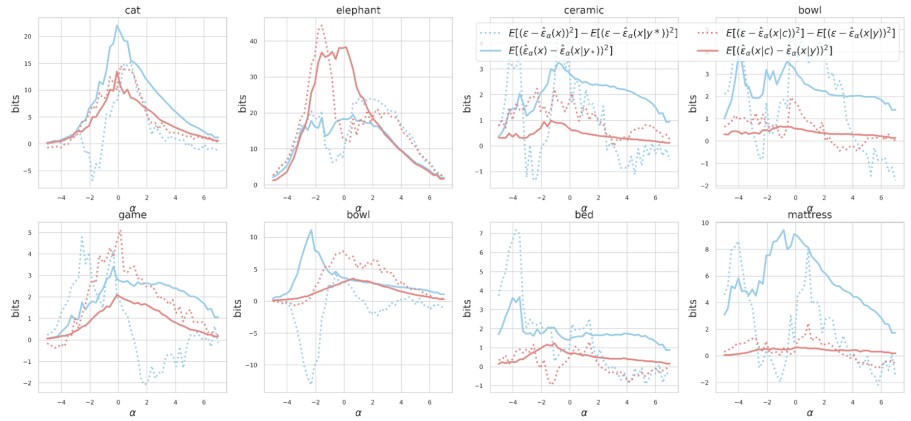

Figure 11: MMSE curves comparing the standard (Eq 2) and orthogonal (Eq 3) estimators

## 6.3 Standard vs Orthogonal Estimators

It can be seen that the original form (dotted line) is more unstable with many zigzag patterns. We also see the orthogonal/simplified form (continuous line) enforces better consistency between the MMSE (blue) and conditional MMSE (red). Thus, this simplification works better in practice.

## 6.4 Estimator Uncertainty

There is no guarantee that the estimator provides an upper or lower bound for the PID terms. It is dependent on the conditional and unconditional denoising MMSEs obtained from the diffusion model which is assumed to be an optimal denoiser for our experiments. However, in practice, this assumption need not hold because a neural network trained on a regression problem to minimize MSE need not converge to the global minima and instead may converge to a local minima. That being said, neural networks have been found to do really well on regression problems and the diffusion model, specifically, has been found to perform well on the denoising problem. Thus, we expect reasonable estimates despite the inherent uncertainty.

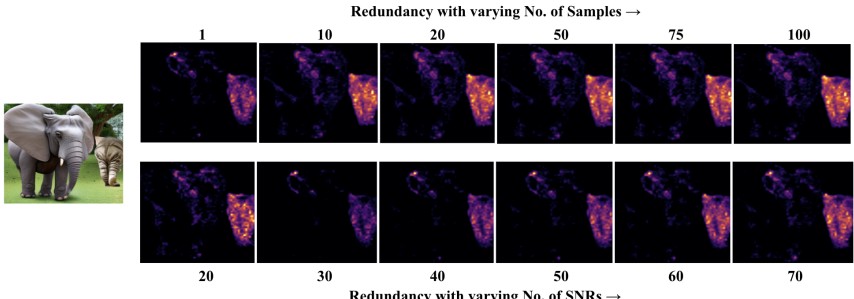

Figure 12: Redundancy maps for the varying levels of noise/SNR and number of samples

A measure of uncertainty is also introduced based on the number of samples under the same noise level, $\alpha$, in Eq 3's expectation term and from the number of values sampled to evaluate the integral in the equation. Thus, we can obtain more confident information maps by using higher values for both of these hyperparameters. In Fig 12, we provide visuals of the information maps for varying values of these hyperparameters on the "cat and elephant" sample from the COCO co-hyponym experiment (Fig 5 in the main text). We observe that the maps depict the same information, i.e., they are highly activated in the same regions across variations but do become less noisy at higher values.

## 6.5 Baselines for Redundancy

When comparing CMI, MI, and DAAM with our redundancy maps, we take the intersection of the maps these methods produce for the two words/phrases individually. This is done in the following steps:

- The individual maps are thresholded to obtain binary masks. The threshold we employ is $\mu + 1.5\sigma$, where $\mu$ refers to the mean of the map being thresholded and $\sigma$ refers to the standard deviation. We use 1.5 times the standard deviation to discard roughly 86.64 % of the map's values as we observe that they are irrelevant and correspond to the low-activated background region.

- We multiply the binary masks of the two words' maps to obtain the final regions of intersection/overlap between the two maps.

- Finally, the above intersection map $im$ is multiplied with each of the two maps, $m_1$ and $m_2$, followed by the computation of the pixel-level mean of the two masked maps to obtain the final map $f$ as:

$$f = (m_1 * im + m_2 * im)/2 \tag{7}$$

## 6.6 Image Editing via Prompt Intervention

We follow a similar process to other image editing methods [79, 80] where we progressively add Gaussian noise to the image and pass it through the diffusion model for denoising but now this reverse diffusion process is conditioned on the edited prompt instead of the original prompt. We edit the original prompt by removing a word from it to see if the edited image is similar to the original to help determine if the omitted word is redundant. For instance, we remove the word "desk" from the prompt "The desk was a sturdy table perfect for working" to produce the edited prompt "The was a sturdy table perfect for working" which is then used to condition the denoising process. The intuition behind this experiment is that for words that contribute little information on top of the rest of the prompt, the model will keep the edited image very similar to the original as it will follow a similar path as that would be followed if it were conditioned on the original prompt.

## 6.7 Datasets

We create datasets for each of our numerous experiments. We primarily rely on existing hierarchical datasets, Wordnet [77] and COCO [76], and free-access Large Language Models, namely Chat-GPT [81], Meta.ai [82], and Gemini [83]. We probe these models with prompts like "*make sentences with pairs of synonyms of common tangible nouns*", "*make sentences with category and subcategory of common tangible nouns*", "*generate sentences with synonyms of common tangible nouns and both of those synonyms must be present in the sentence*", "*generate pairs of sentences containing homonyms in different contexts*" and so on. A combination of the outputs produced by these models was used to obtain the final set of prompts for each task. Once we had the prompts, we fed them into the open-source diffusion model, Stable Diffusion 2.1 [7], to generate images corresponding to those prompts. During the image generation, we kept the seed constant (42). Following are the prompt-image pair datasets we introduce:

- **Gender Bias Dataset**: Here we generated a list of 188 most common occupations from the LLMs ([82],[81],[83]). These occupations are then combined with each gender one at a time to produce a total of 376 prompts. For instance, the occupation "Doctor" is used to generate two prompts, "Female Doctor" and "Male Doctor". A sample of this dataset is visible in Fig 13.

- **Ethinic Bias Dataset**: Instead of combining the previously mentioned 188 occupations with a gender, we now combine them with one of four ethnicities (*Caucasian, Asian, Black, Hispanic*) in each prompt to produce a total of 752 prompts. For example, the occupation "Doctor" is used to generate four prompts, "Caucasian Doctor", "Asian Doctor", "Black Doctor", and "Hispanic Doctor". A sample of this dataset is visible in Fig 14.

- **Homonym Dataset:** For this dataset, we generated 121 pairs of prompts, where each pair corresponds to one homonym but in different contexts. Thus, in total, we produced 242

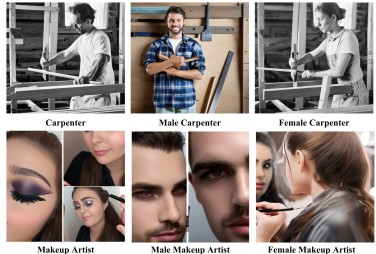

Figure 13: Gender Bias Dataset Sample

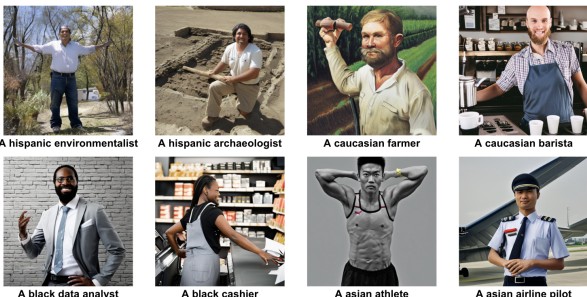

Figure 14: Ethnic Bias Dataset Sample

prompts, all of which were generated from the LLMs [82],[81],[83]. A sample of this dataset is visible in figures 15a.

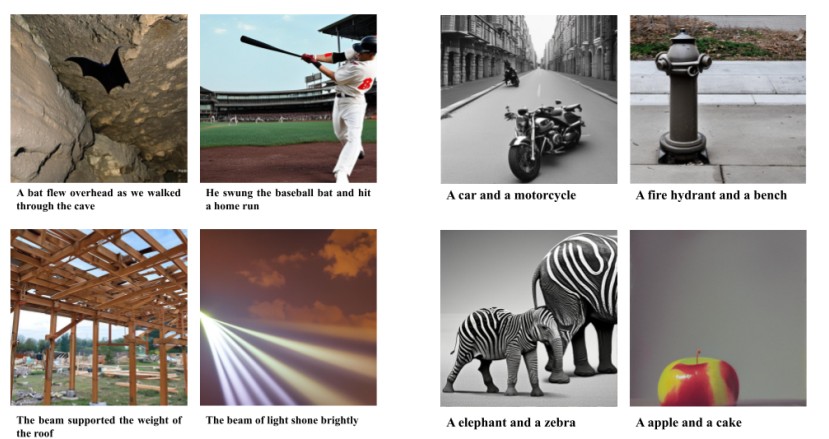

(a) Homonym Dataset Sample  (b) Co-Hyponym Dataset (COCO) Sample

- **Synonym Dataset:** The entire dataset of 132 prompts containing synonym pairs was generated using the previously mentioned LLMs [82],[81],[83]. A sample of this dataset is visible in Fig 16.

- **Co-Hyponym Dataset (Wordnet)**: To create this dataset, we first sampled co-hyponyms from Wordnet for the most frequently occurring tangible nouns in the COCO dataset. Next, we generated prompts with the format: "a *co-hyponym* and a *co-hyponym*". For example, one of the most commonly used nouns across the COCO dataset is "bench", so we find its co-hyponyms from Wordnet: "box", "seat", "chair", "ottoman", "stool", "sofa" and "couch". Then, these are used to produce prompts such as "a box and a bench", "a chair and a bench", "a sofa and a bench" and so on. A sample of this dataset is visible in Fig 17.

- **Co-Hyponym Dataset (COCO)**: We took the super-categories of the COCO dataset and paired the objects within each individual super-category to produce a list of 197 co-hyponym

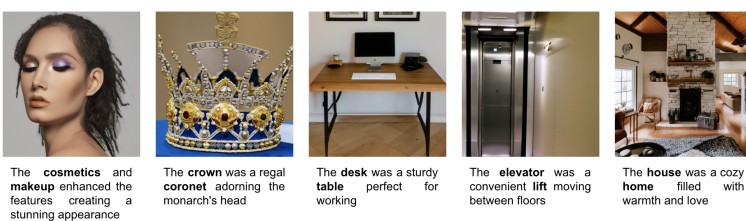

The **cosmetics** and **makeup** enhanced the features creating a stunning appearance

The **crown** was a regal **coronet** adorning the monarch's head

The **desk** was a sturdy **table** perfect for working

The **elevator** was a convenient **lift** moving between floors

The **house** was a cozy **home** filled with warmth and love

Figure 16: Synonym Dataset Sample

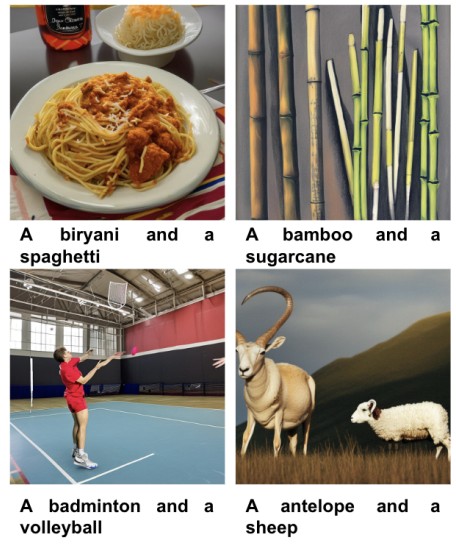

A biryani and a spaghetti

A bamboo and a sugarcane

A badminton and a volleyball

A antelope and a sheep

Figure 17: Wordnet Dataset Sample

pairs which are then converted to prompts using a similar format as that used for the Wordnet dataset above. For example, "vehicle" was a super-category, so "a car and a airplane", "a car and a boat" and "a motorcycle and a truck" are a few prompts produced from that super-category. A sample of this dataset is visible in Fig 15b.

## 6.8 More Results

In this section, we present more results from our experiments.

### 6.8.1 Homonyms

Below are some more examples of results for Homonym Experiments.

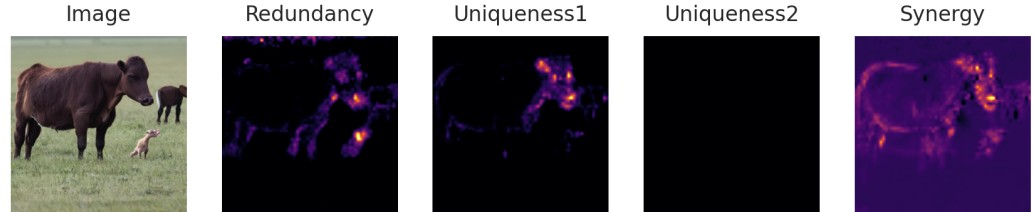

c = the calf frolicked in the field alongside its mother
y = 'calf' vs 'field'

| Image | Redundancy | Uniqueness1 | Uniqueness2 | Synergy |
|---|---|---|---|---|

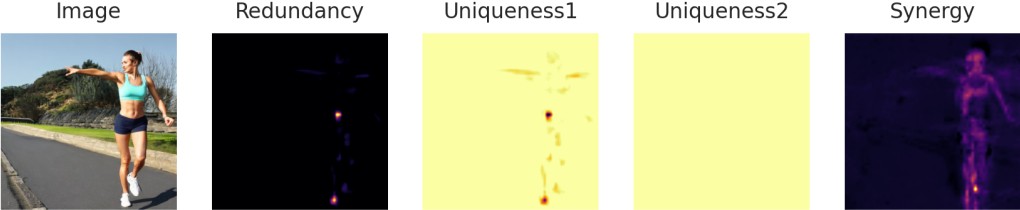

c = she stretched her calf muscles before going for a run
y = 'calf' vs 'run'

| Image | Redundancy | Uniqueness1 | Uniqueness2 | Synergy |
|---|---|---|---|---|



c = she admired the stained glass windows in the church
y = 'glass' vs 'windows'

| Image | Redundancy | Uniqueness1 | Uniqueness2 | Synergy |
|---|---|---|---|---|

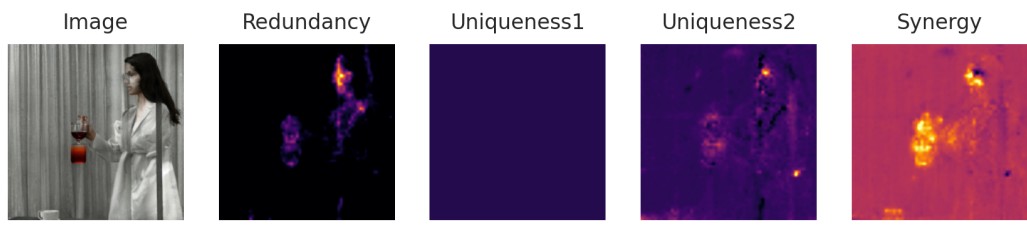

c = she sipped her water from a tall glass
y = 'glass' vs 'water'

| Image | Redundancy | Uniqueness1 | Uniqueness2 | Synergy |
|---|---|---|---|---|

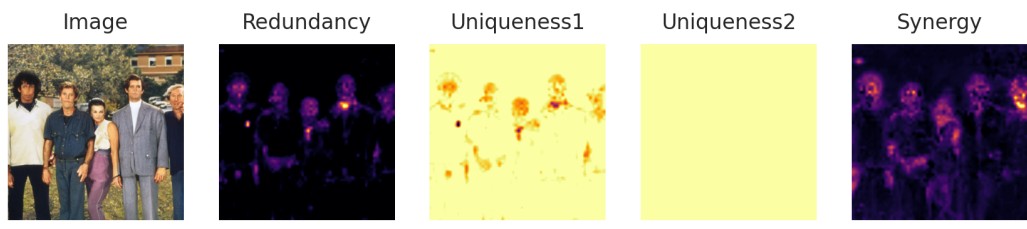

c = the gang worked together
y = 'gang' vs 'together'

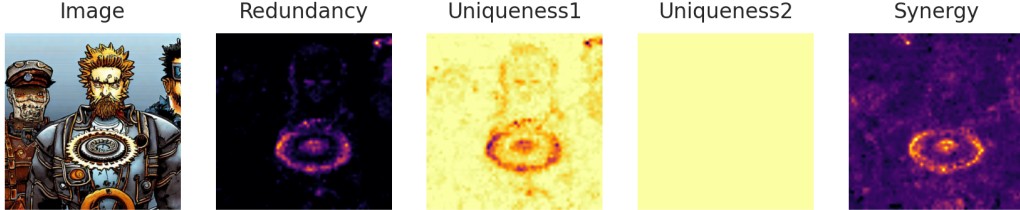

| Image | Redundancy | Uniqueness1 | Uniqueness2 | Synergy |
|-------|-----------|-------------|-------------|---------|

c = the gang of gears was complex
y = 'gang' vs 'gears'

.

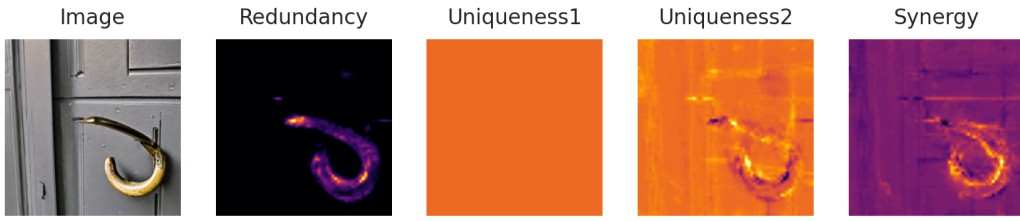

| Image | Redundancy | Uniqueness1 | Uniqueness2 | Synergy |
|-------|-----------|-------------|-------------|---------|

c = the hook on the door was useful
y = 'hook' vs 'door'



| Image | Redundancy | Uniqueness1 | Uniqueness2 | Synergy |
|-------|-----------|-------------|-------------|---------|

c = he baited the hook with a worm
y = 'hook' vs 'worm'

Below are some Synergy map visuals obtained from CPID on Homonym samples.

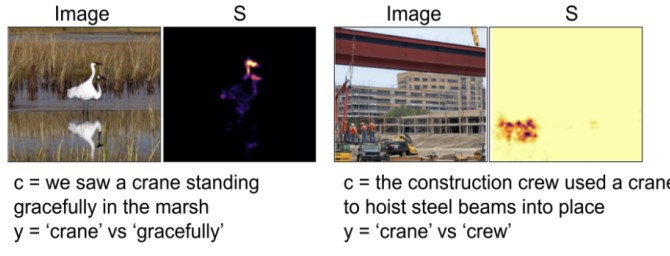

c = we saw a crane standing
gracefully in the marsh
y = 'crane' vs 'gracefully'

c = the construction crew used a crane
to hoist steel beams into place
y = 'crane' vs 'crew'

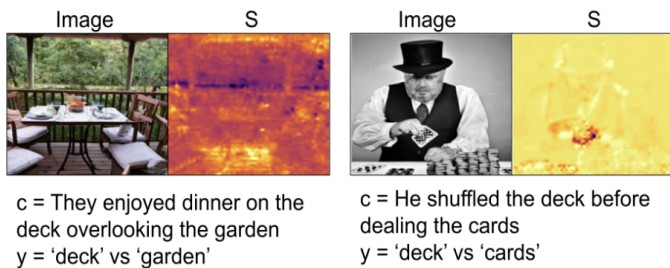

c = They enjoyed dinner on the deck overlooking the garden
y = 'deck' vs 'garden'

c = He shuffled the deck before dealing the cards
y = 'deck' vs 'cards'

### 6.8.2 Synonyms

Below are some more examples of results for Synonyms Experiments.

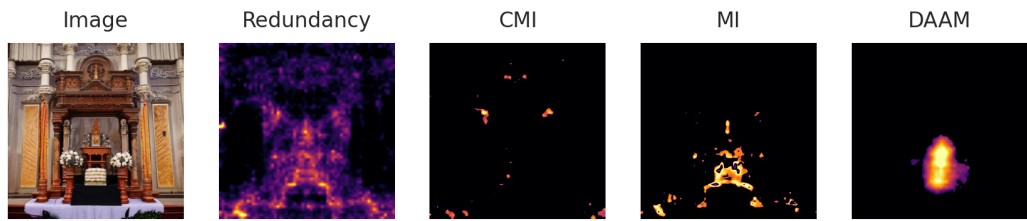

c = the altar was adorned with a beautiful altar-piece adorned with intricate carvings
y = 'altar' vs 'altar-piece'

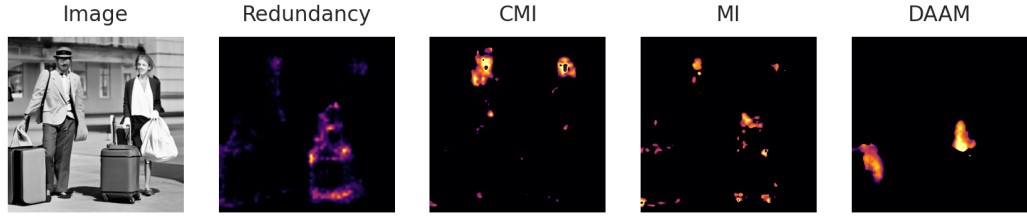

c = the luggage and bag carried belongings with ease making travel a breeze
y = 'luggage' vs 'belongings'

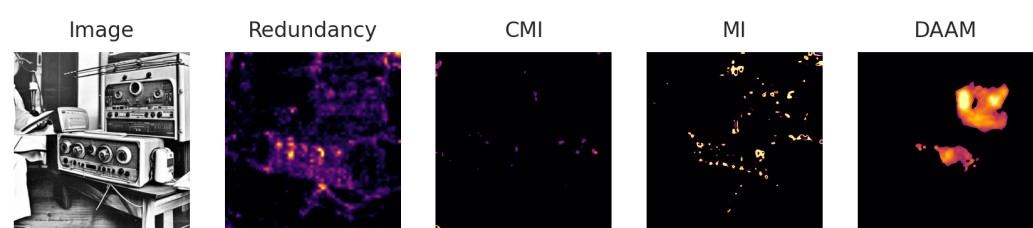

c = the radio was a broadcasting transmitter playing music and news
y = 'radio' vs 'transmitter'

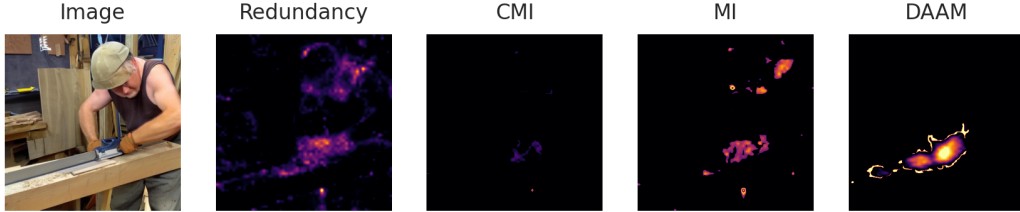

c = the saw was a sharp tool cutting through wood easily
y = 'saw' vs 'tool'

### 6.8.3 COCO Co-Hyponyms

Below are the results of Co-Hyponyms.

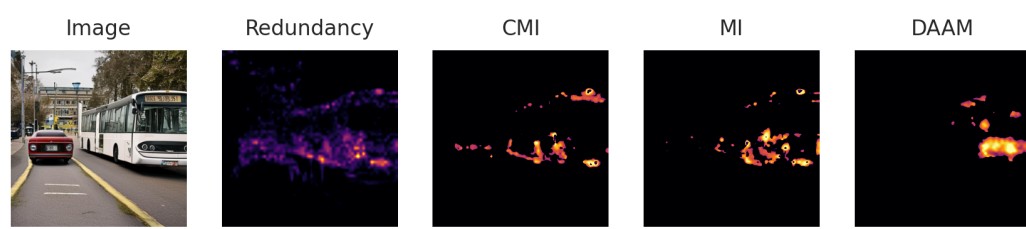

c = a car and a bus
y = 'car' vs 'bus'

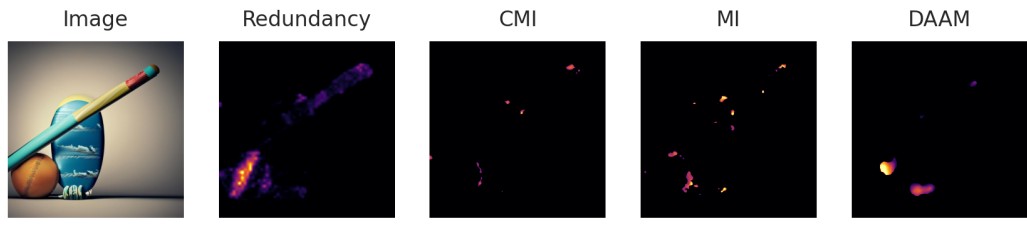

c = a baseball bat and a surfboard
y = 'baseball' vs 'surfboard'

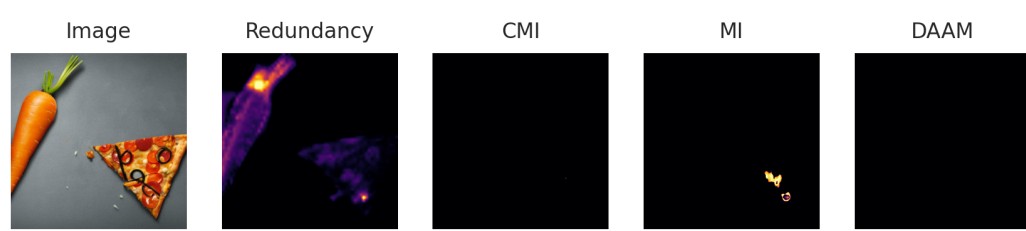

c = a carrot and a pizza
y = 'carrot' vs 'pizza'

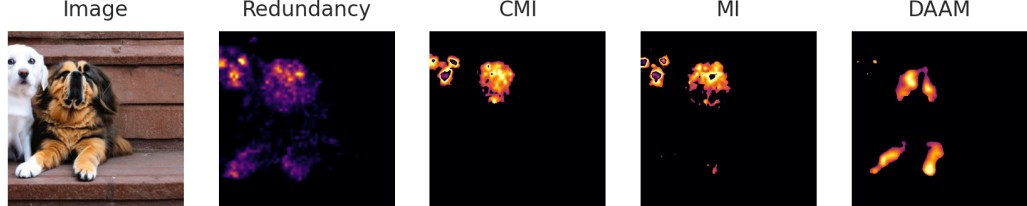

c = a cat and a dog
y = 'cat' vs 'dog'

### 6.8.4 Wordnet Co-Hyponyms

Below are some more results for the Co-Hyponym experiments on the images generated from prompts made through Wordnet.

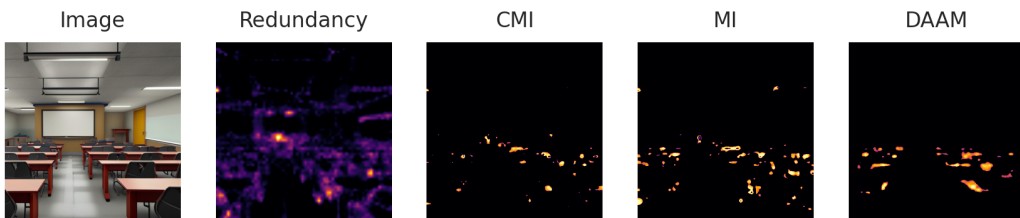

c = a chamber and a classroom
y = 'chamber' vs 'classroom'

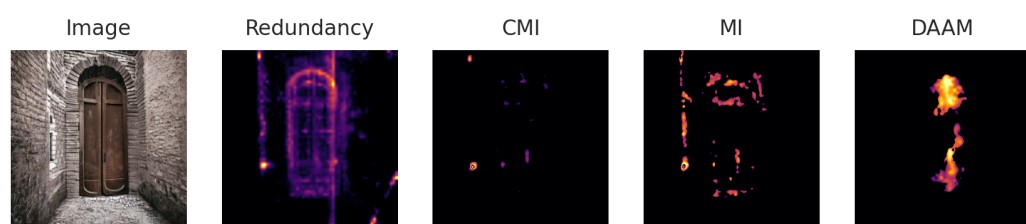

c = a door and a tower
y = 'door' vs 'tower'

| Image | Redundancy | CMI | MI | DAAM |
|-------|-----------|-----|-----|------|

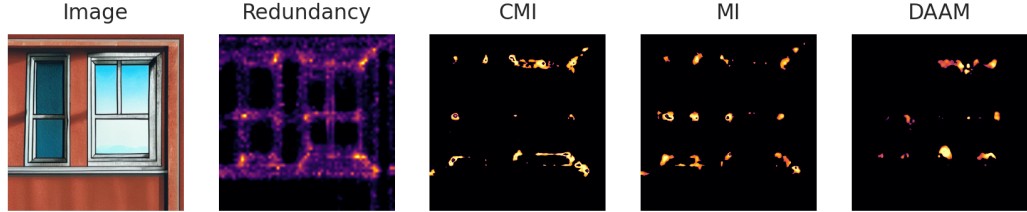

c = a glass and a window
y = 'glass' vs 'window'

| Image | Redundancy | CMI | MI | DAAM |
|-------|-----------|-----|-----|------|

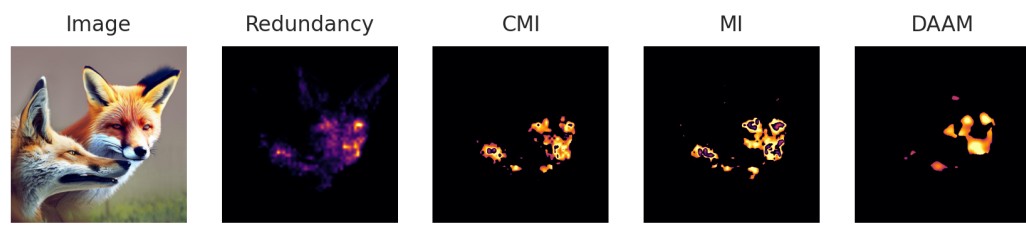

c = a fox and a dog
y = 'fox' vs 'dog'

### 6.8.5 Prompt Intervention

Below are some more results for the Prompt Intervention experiments.

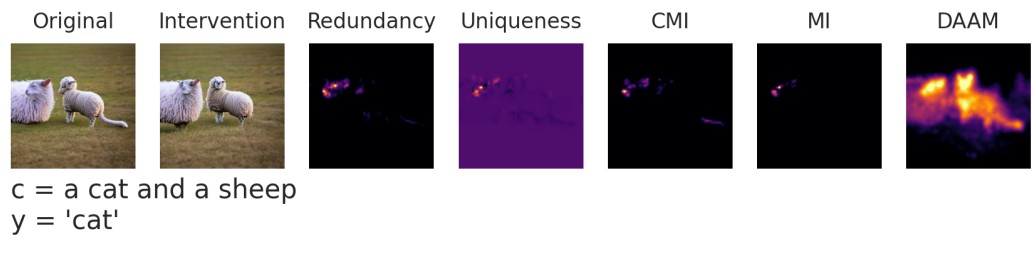

c = a cat and a sheep
y = 'cat'

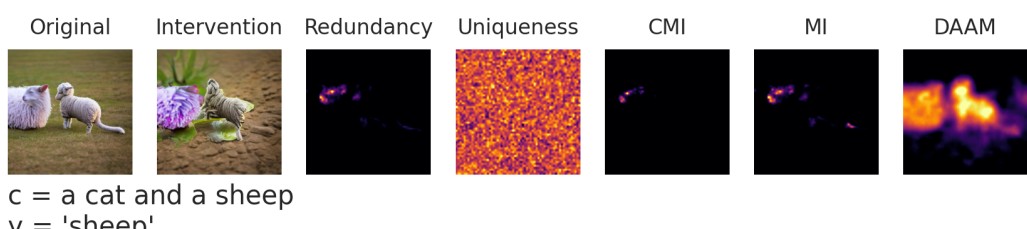

c = a cat and a sheep
y = 'sheep'

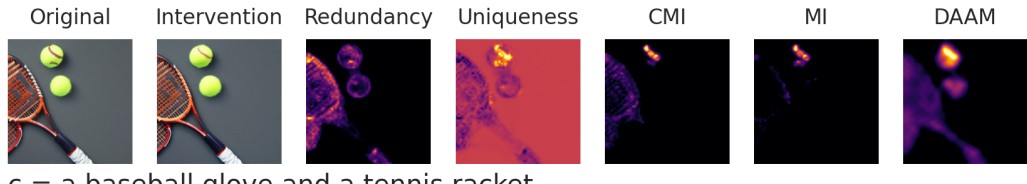

c = a baseball glove and a tennis racket
y = 'baseball'

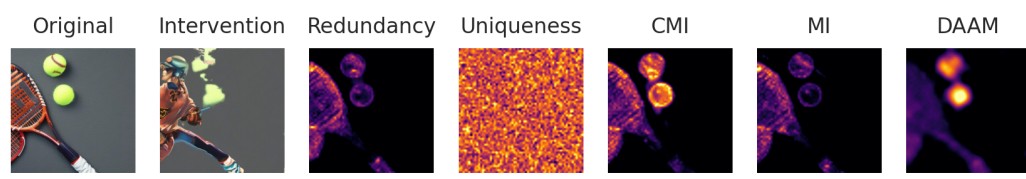

c = a baseball glove and a tennis racket
y = 'tennis'

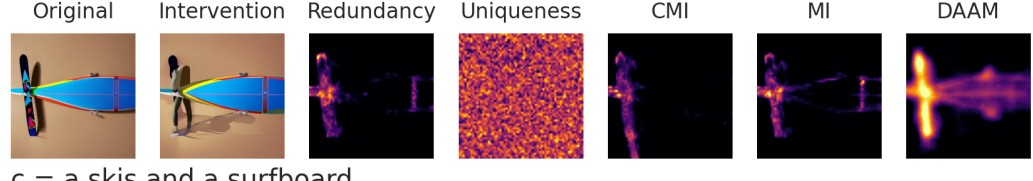

c = a skis and a surfboard
y = 'skis'

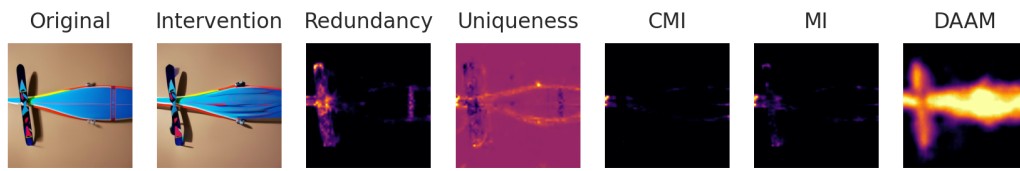

c = a skis and a surfboard
y = 'surfboard'

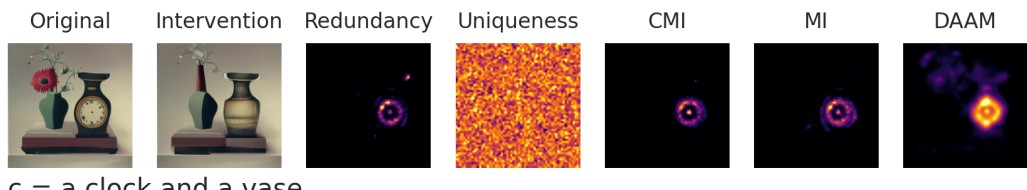

c = a clock and a vase
y = 'clock'

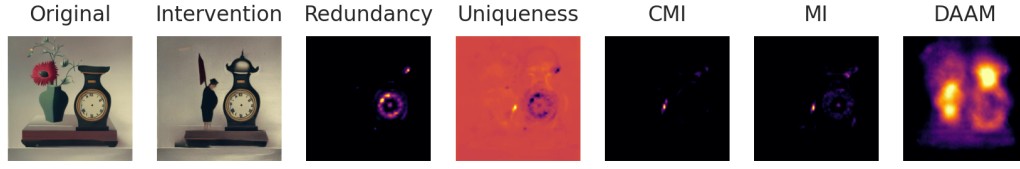

c = a clock and a vase
y = 'vase'

