# OpenReview forum: "Diffusion PID: Interpreting Diffusion via Partial Information Decomposition"
_NeurIPS.cc/2024/Conference — NeurIPS 2024 poster_

### Official Review · Reviewer_THqw · 2024-06-23

**Soundness:** 3
**Presentation:** 3
**Contribution:** 4
**Rating:** 6
**Confidence:** 4

**Summary:**

In this paper,  the authors propose a novel approach to analyze the uniqueness, redundancy, and synergy terms in text-to-image diffusion models by applying information-theoretic principles to decompose the input text into its elementary components. In particular, the proposed approach can be used to recover gender and ethnicity biases in image generation.

**Strengths:**

1.	It is an important and valuable direction to analyze the potential semantic bias in the field of image generation. I think it is also reasonable to invovle information-theoretic principles into the approach.

2.	The proposed approach is a good attempt for measuring the gender and ethnicity biases in image generation.

3.	Based on the illustration in Figure 1-3, the proposed approach may provide more interpretability for text-to-image diffusion models, which is important to understand and exploit the models.

**Weaknesses:**

1.	Are there something wrong in Equation 2? The left parts in those two equations are the same but the right parts are not.

2.	The time complexity if necessary for the proposed method

**Questions:**

1.	Please provide more analysis on the efficiency of the proposed approach.

2.	Actually, I understand that it is not easy to evaluate the effectiveness and make others convincing with only some case studies in Figure 1-3. It is better to publish the demo, where readers can test the approach by themselves.

---

> ### Author Rebuttal · Authors · 2024-08-07
>
> We thank you for your valuable feedback and helpful comments. We address your concerns below.
>
> > **Q1)** Clarification on Eq 2:
>
> We thank you for mentioning this. Unfortunately, our footer note didn't make it through the submission. Eq 2 is correct, and the second equation is derived from the first using the orthogonality principle, as explained below:
>
>
>
> $I(X; Y) = \mathbb{E}_{p(x,y)} [\log p(x|y) - \log p(x)]$
>
>
>
> $I(X; Y) = \mathbb{E}_{p(x,y)} [ \frac{1}{2}  \int \mathbb{E}\_{p(\epsilon)}[ \|\| \epsilon - \hat{\epsilon}\_{\alpha}(x\_{\alpha})\|\| ^2 - \
>  \| |\epsilon - \hat{\epsilon}\_\alpha (x\_{\alpha} | y) \| \|^2 ] d\alpha]$
>
> By expanding all the squares and re-arranging we get:
>
> $I(X; Y) = \mathbb{E}\_{p(x,y)} \left[ \frac{1}{2} \int \mathbb{E}\_{p(\epsilon)} \left[ \| \| \hat{\epsilon}\_\alpha (x\_\alpha) - \hat{\epsilon}\_\alpha (x\_\alpha | y) \| \|^2 \right] d\alpha \right] + 2 \mathbb{E}\_{p(y)} \left[ \frac{1}{2} \int \mathbb{E}\_{p(x|y), p(\epsilon)} \left[ (\hat{\epsilon}\_\alpha (x\_\alpha) - \hat{\epsilon}\_\alpha (x\_\alpha | y)) \cdot (\hat{\epsilon}\_\alpha (x\_\alpha | y) - \epsilon) \right] d\alpha \right]$
>
> Here,
>
> $+ 2 \mathbb{E}\_{p(y)} \left[ \frac{1}{2} \int \mathbb{E}\_{p(x|y), p(\epsilon)} \left[ (\hat{\epsilon}\_\alpha (x\_\alpha) - \hat{\epsilon}\_\alpha (x\_\alpha | y)) \cdot (\hat{\epsilon}\_\alpha (x\_\alpha | y) - \epsilon) \right] d\alpha \right] \equiv \ominus$
>
> based on the orthogonality principle [1], which states:
>
> $\forall f, \quad \mathbb{E}\_{p(x|y) p(\epsilon)} \left[ f(x\_\alpha, y) \cdot (\hat{\epsilon}\_\alpha (x\_\alpha | y) - \epsilon) \right] = 0$
>
> The term $(\hat{\epsilon}\_\alpha (x\_\alpha | y) - \epsilon)$ represents the error of the MMSE estimator, which is orthogonal to any estimator $f$. Therefore, the second term becomes zero, leading to:
>
> $I(X; Y) = \mathbb{E}\_{p(x,y)} \left[ \frac{1}{2} \int \mathbb{E}\_{p(\epsilon)} \left[ \| \| \hat{\epsilon}\_\alpha (x\_\alpha) - \hat{\epsilon}\_\alpha (x\_\alpha | y) \| \|^2 \right] d\alpha \right]$
> which is the 2nd line in Eq 2.
>
>
> **Fig 6: MMSE curves comparing the standard and orthogonal estimators**
>
> We also provide graphs comparing the MMSE estimate obtained from the original equation form (the first equation in Eq 2) and the simplified form for varying levels of noise/SNR in **Fig 6** in the attached PDF. It can be seen that the original form (dotted line) is more unstable with many zigzag patterns. We also see the orthogonal/simplified form (continuous line) enforces better consistency between the MMSE (blue) and conditional MMSE (red). Thus, this simplification works better in practice.
>
>
> > **Q3)** Time analysis
>
> Our primary goal in this work was to help improve the interpretability of diffusion models. Although the time complexity of such methods is not of concern usually given that they are used for interpretability of a model, we understand that this is an important aspect. Methodologically, once we have the generated image from diffusion, our method involves running the diffusion model's UNet for denoising to get the MMSE-based log probability estimates and BERT to obtain the terms' probabilities, as required for PID. Thus, our model's time complexity is proportional to that of the diffusion + BERT model. The exact time would depend on the hyperparameters and the computational resources used.
>
>
>
>
> > **Q4)** Public demo and code release
>
> We would like to clarify that we provide code for our method in the supplemental zip, which is visible to the reviewers. This code release will help ensure reproducibility, soundness check and facilitate future research.
>
> We are unable to share a hosted model at this current point in time as we need to maintain anonymity as per the guidelines. We will also release the code and models publicly after the end of the anonymity period. We will also add a public demo for easy usage and testing of our pipeline.
>
>
>
>
> [1] Steven M. Kay. 1993. Fundamentals of statistical signal processing: estimation theory. Prentice-Hall, Inc., USA.

---

> > ### Comment · Reviewer_THqw · 2024-08-13
> >
> > Thanks for your responses.  I'm glad to improve my score.

---

> > > ### Author Response · Authors · 2024-08-13
> > >
> > > We are incredibly grateful for your generous assessment of our work! Your positive feedback means a great deal to us, and we will be sure to implement the necessary changes in the next revision.
> > >
> > > Thank you once again for your valuable insights and support.

---

### Official Review · Reviewer_hzC5 · 2024-07-10

**Soundness:** 3
**Presentation:** 4
**Contribution:** 3
**Rating:** 7
**Confidence:** 5

**Summary:**

Decomposing different types of information (redundant, synergistic, unique) has long been a niche field due to the difficulty of applying these methods in realistic settings. This paper develops a way to dissect these types of fine-grained information measures in a practical way in the realistic setting of text-to-image diffusion models. The authors perform extensive experiments to show that the resulting method can be useful for identifying biases, understanding model errors, probing complex linguistic features, and for model interpretability in general.

**Strengths:**

- Well-written motivation and examples
- The adaptation of PID to a tractable diffusion model measure looked elegant, and included some nontrivial twists (including accounting for the pointwise measure ambiguities)
- Good overview of related work including broader attempts at model interpretability
- Extensive experiments explore a wide array of interesting
- The provided datasets can be a benefit to future research.
- In the cases where you could quantify biases, it was nice to see that you get a large and reliable signal.

**Weaknesses:**

- Interpretability research is intrinsically very challenging, as it is hard to verify ground truth and even saying what counts as "interpretable" can be a bit nebulous. Generally the information maps looked intuitive, but in some cases I felt I had to squint a little to see the relationship.
- Related to that, there's some uncertainty in your estimator (terms in Eq. 2). It would be nice to understand how noisy / confident the information maps are.
- Comparisons can also be difficult - there's no existing method that exactly does what PID does, and this is the first approach to extend PID to high-dimensions. The comparisons with MI/CMI and attention methods like DAAM seemed like a reasonable way to handle this.
- In some cases, it was hard to imagine how we could use the results. For instance, consider the homonym failure case identified by synergy. To make use of this in practice, the user would have to know to associate a homonym with a specific context word that should exhibit synergy. This is sort of an exploratory work, but finding more concrete applications would improve impact.

**Questions:**

For PID veterans, Eq. 5 is straightforward. Depending on other reviewers' reactions, it might be necessary to give more context (like the famous diagram that is often associated with it).

Your PID was heavily based on Williams and Beer measure (though adapted for the pointwise measure). I'm curious, did you consider other PID formulations besides Beer & Williams? Probably not something that needs to be addressed in the paper since NeurIPS community is not very familiar with these ideas (yet).

I haven't studied this paper, but I know other people have thought about how to adapt PID to pointwise measures also:
Finn C, Lizier JT. Pointwise Partial Information Decomposition Using the Specificity and Ambiguity Lattices.

Minor formatting quibble - log and arg in Eq 1,2 should be formatted as operators, not italics.

**Limitations:**

This was discussed adequately.

---

> ### Author Rebuttal · Authors · 2024-08-07
>
> We thank you for your comprehensive review and thoughtful comments. We also appreciate your comments for explicitly highlighting the difficulties of adapting PID to diffusion and the challenges of working on the interpretability of models.
>
> > **Q1)** Relationship between PID maps and human intuition
>
> We thank you for acknowledging and drawing attention to the hurdles in this kind of work. Although we show examples that align with human intuition, we also show examples where it's possible that not every relation matches human intuition, as we are exploring the diffusion model's understanding of the world. We also introduce CPID (Conditional PID) as an extension of PID to better account for the context provided by the rest of the prompt. We refer you to the global response above for further details on CPID and the attached PDF for visuals (**Figs 1, 2, 3, 4**).
>
> > **Q2)** Estimator's uncertainty
>
> We expand further on this query in the global response above and will also add an explanation in the paper's next revision.
>
> > **Q3)** Applications of DiffusionPID
>
> The primary goal of our work was to introduce the concept of PID into the rapidly growing generative field in computer vision. We believe this would lay down the groundwork to further the understanding of the internal concepts learned by diffusion. This would produce more concrete ways to evaluate and progress prompt engineering. PID can identify which elements of a prompt provide unique information, allowing for the refinement of prompts to reduce redundancy and enhance synergy. Our method can also help better understand and counter diffusion's attribute binding problems and learned biases such as those demonstrated in our paper. We also believe incorporating PID-based insights into model training could ensure that the model better captures human-like understanding of concepts and thus make it more human-aligned.
> One way to address the point regarding the requirement of apriori knowledge of which words to run PID on is that several pairs of terms could be sampled automatically from the prompt (with some filtering of pairs such as based on LLM-based semantic distance between the terms) and run through PID. We agree that the practical application of PID requires further work and is something we plan to explore in the future but believe that this work provides the necessary foundation for this direction.
>
> > **Q4)** PID Diagram
>
> Thank you for raising this point. We have added a figure (**Fig 5**) to the attached PDF to give more insight into the concept.
>
> > **Q5)** Williams and Beer and Finn C, Lizier JT's work
>
> Yes, as you rightly recognized, one of the major reasons behind building on the Williams and Beer formulation was the pointwise measure. The choice was also based on the observation that the NeurIPS and the AI community previously used this formulation [1]. Furthermore, both works were quite helpful for our paper. We thank you for mentioning the latter and will make sure to cite it in the next revision of the paper.
>
> > **Q6)** Formating and typos
>
> Thank you for bringing this to our attention and we will fix this in the next revision of the paper.
>
> [1] Yu, S., Wickstrøm, K., Jenssen, R., & Principe, J. C. (2020). Understanding convolutional neural networks with information theory: An initial exploration. IEEE transactions on neural networks and learning systems, 32(1), 435-442.

---

> > ### Comment · Reviewer_hzC5 · 2024-08-09
> > **Response**
> >
> > Thanks, for the detailed response.
> > I read the reviews and rebuttals and maintain my score.

---

> ### Author Response · Authors · 2024-08-09
>
> We are deeply grateful for the positive assessment you have given to our work! Will make the required changes in the next revision of the paper.
>
> Thank you again for your suggestions and comments.

---

### Official Review · Reviewer_Tpva · 2024-07-11

**Soundness:** 3
**Presentation:** 3
**Contribution:** 2
**Rating:** 6
**Confidence:** 3

**Summary:**

The paper proposes a new technique called DiffusionPID to explain how diffusion models transform text cues into images through partial information decomposition (PID). This work deconstructs mutual information into redundancy, synergy, and uniqueness to analyze how individual concepts and their interactions shape the generated images. This paper conducts extensive experiments and visualizations to validate its techniques.

**Strengths:**

1.	This paper proposes new techniques to explain the influence of multiple concepts and their interactions on diffusion models.
2.	This paper is well-written and easy to follow.
3.	There are extensive visualizations for many scenarios to interpret the process of diffusion models. This work explains some problems, such as bias and incorrect generation, based on its method, which indicates the direction for improvement.

**Weaknesses:**

1.	The experiments primarily focus on the impact of interactions among multiple object concepts on image generation, lacking analysis of more general scenarios, such as the interactions between objects and their attributes.

**Questions:**

1.	Why do you use BERT to measure the p(y)? Does the choice of different encoders significantly impact p(y)?

**Limitations:**

1.	The analysis lacks exploration of more complex text prompt scenarios, such as those involving interactions among more objects and more complex phrases that encompass attributes along with objects.

---

> ### Author Rebuttal · Authors · 2024-08-07
>
> We thank you for your thorough review and insightful feedback. We address your suggestions below and also refer to the global response at the top in a few places.
>
> > **Q1)** Analysis on objects and attributes
>
> We agree that our work could be further improved by providing additional analysis on the interaction between objects and their attributes. We address this by providing new visuals for these types of interactions in **Figs 1 and 2** in the attached PDF along with other experiments (CPID) as detailed in the global response above. We will also include these figures in the next revision of the paper.
>
>
> > **Q2)** Choice of BERT
>
> We would like to point out that BERT was the original language backbone of choice for latent diffusion [1]. They emperically find that BERT can be effectively used to encode semantic information of images in a generation setting.
> Furthermore, the common methods to compute $p(y)$ involve modeling a distribution of the language's vocabulary in natural language. For this, the only options are: textual databases or language models.
> Online available databases (ex. Wikipedia:Word frequency, Google Books Ngram Viewer, Corpus of Contemporary American English, etc.), can only be used to obtain the independent probability of each term based on frequency statistics but it remains difficult to obtain an accurate conditional distribution. Moreover, to calculate this over large corpuses online would be computationally very expensive. That's why we used a language model instead.
> BERT has been the standard language model used in natural language research since its conception due to its excellent performance on various language-based tasks and its accurate model of the vocabulary distribution in both, conditioned and unconditioned cases. We did not feel the need to use LLMs as the context window here is fairly small given that we operate on prompts of moderate length. Thus, this choice does not significantly impact $p(y)$.
>
>
>
> [1] Robin Rombach, Andreas Blattmann, Dominik Lorenz, Patrick Esser, and Björn Ommer. Highresolution image synthesis with latent diffusion models. In Proceedings of the IEEE/CVF conference on computer vision and pattern recognition, pages 10684–10695, 2022.

---

> > ### Comment · Reviewer_Tpva · 2024-08-13
> > **Thanks for your response.**
> >
> > Thanks for the detailed response, I'm glad to keep my positive rating.

---

> > > ### Author Response · Authors · 2024-08-13
> > >
> > > We truly appreciate your positive evaluation of our work! We will make the necessary adjustments in the next revision of the paper.
> > >
> > > Thank you once again for your insightful feedback.

---

### Official Review · Reviewer_vbX2 · 2024-07-12

**Soundness:** 2
**Presentation:** 3
**Contribution:** 4
**Rating:** 6
**Confidence:** 5

**Summary:**

Summary:
The paper adapt the concept of the Partial Information Decomposition into the diffusion model and analyze the uniqueness, redundancy and synergy terms in the diffusion model and do experiments on the Bias, Homonyms and Synonyms

Contribution: The paper adapt the concept of the Partial Information Decomposition into the diffusion model and out-perform other method s in visual perspective and explain the bias homonym and synonyms in the text token perspective by using the PID

**Strengths:**

Strength:

(i) Clarity: The paper is easy to understand.

(ii) Originality and Significance: The paper is the first one to adapt the concept of the PID into diffusion model and consider the influence of more than one token on the generated images. It also help explained the  Bias, Homonyms and Synonyms problem in the diffusion model compared to other current methods.

**Weaknesses:**

Weaknesses:

(i) The paper's novelty is limited. The paper does not go beyond the concept of PID but simply adapt the PID concept to the diffusion model. Maybe extend to more than two tokens.


(ii) The text conditions is simple. Try the text conditions that more similar to the distribution of the training data.


(iii) The uniqueness figures in the 6.5.1 is not very convincing to me and not show very clear information.

**Questions:**

(i) I think the text condition  that you choose to present can be more complicated which may present more issue when the user's prompt are complicated.


(ii) Why some of the uniqueness figures in the 6.5.1 Homonyms sections show meaningless information? How is the uniqueness different from other single token method?  Is it more convincing?

**Limitations:**

Yes

---

> ### Author Rebuttal · Authors · 2024-08-07
>
> We thank you for your detailed review and thoughtful comments. We address your concerns below and will refer to the general response at the top for certain points.
>
> > **Q1)** Novelty of our method and more complex use cases such as an extension to the multi-concept scenario
>
> 1. Our approach is the first to adapt PID for diffusion models by developing the required mathematical formulations. We derive a mathematically sound pixel-level PID breakdown in a form that is compatible with and could be integrated into diffusion. Furthermore, we leverage our method to conduct a detailed study of diffusion and to identify its various failure modes, which significantly advances our understanding of different concepts and their interactions in the diffusion model.
> 2. Building upon the original concept of PID, we also introduce Conditional PID (CPID) for diffusion models to expand the scope and contributions of this work and provide more details in the global response above (**Figs 1, 2, 3, 4** in the attached PDF). This also helps address the suggestion on expanding to more terms as CPID takes the rest of the prompt into consideration. We observed that CPID yields superior localized results compared to its PID counterpart in practice as it takes into account the rest of the prompt as context during the image generation process, allowing it to better capture the specific contributions of the two tokens being analyzed. We will include the results in the next revision of the paper as well.
> 3. Computing the PID and even MI (Mutual Information) between three or more input variables in information theory is a complex process. There is no universally accepted method for defining and calculating the terms of Uniqueness, Redundacy and Synergy. While works such as [1] provide a method to compute the global Redundancy between all input variables, Uniqueness and Synergy remain difficult terms to tackle as these computations need to take all the variables subsets' interactions into consideration. It is generally agreed [1, 2] that multivariate information decomposition is more involved than the bivariate case because it is not immediately obvious how many atoms of information one needs to consider, nor is it clear how these atoms should relate to each other. Even the breakdown provided in [2] for the four-variable case is highly complicated and it is a non-trivial problem to extend these concepts from information theory to more variables.
> 4. The primary objective of the analysis conducted in the main text was to learn more about the text-to-image diffusion model's understanding of different concepts, which may or may not align with human understanding, and to derive an analysis for specific prompt types. We agree that the examples used in the paper are relatively simple, but they can still shed light on the various interactions and shortcomings of the diffusion model in an easily interpretable form. To address the need for more complex prompts, we provide results on longer prompts with more entities, similar to those used in the diffusion training data, in **Figs 1 and 2** in the attached PDF and provide more details on the new experiments in the global response above. We will include the new figures in the next revision of the paper as well.
>
> > **Q2)** Clarification on uniqueness figure 6.5.1 and distinction from single-concept methods
>
> The uniqueness information maps in Fig 6.5.1 tries to show what information that term uniquely contributes on its own to the image generation process. Single-concept methods like DAAM, MI, and CMI highlight the overall information a concept contributes which is a combination of Synergy, Redundancy and Uniqueness. We clarify this in Eq 5 in the main text. It is possible that something can have little Uniqueness, i.e., contributes very little unique information on its own given the rest of the prompt or even just another concept as sufficient context, but still have a contribution through Synergy and/or Redundancy. For instance, in the "calf" vs "field" example, the concept "calf" when interpreted as the calf of an animal, is enough to make the model generate a field. Thus, the "field" concept does not contribute much unique information on its own but it does provide synergistic information in combination with "calf" that helps the model interpret the meaning of "calf" as the animal and not the muscle, which is why we see a high Synergy. As long as there is some other term instead of "field" that can provide the required context to make the diffusion interpret the "calf" as the animal, then removing "field" would still generate the grass field in the image. Similar arguments can be made for the rest of the examples as well. Also, our approach does not change depending on the order of the concepts being processed by DiffusionPID. We will update the figure captions in an upcoming revision to clarify this point. We thank you for bringing this to our attention.
>
>
> [1] Conor Finn and Joseph T. Lizier. Pointwise information decomposition using the specificity and ambiguity lattices. ArXiv, abs/1801.09010, 2017.
>
> [2] Paul L. Williams and Randall D. Beer. Nonnegative decomposition of multivariate information, 2010.

---

> > ### Comment · Reviewer_vbX2 · 2024-08-12
> >
> > Thanks for addressing my concerns, I decided to maintain my score for now.

---

> > > ### Author Response · Authors · 2024-08-13
> > >
> > > We sincerely appreciate your positive feedback on our work! We will incorporate the necessary changes in the next revision of the paper.
> > >
> > > Thank you so much for your valuable suggestions and comments.

---

### Author Rebuttal · Authors · 2024-08-07

We thank all the reviewers for taking the time to go through our work in detail and providing insightful reviews. We found the feedback very constructive and helpful.

We are glad that the reviewers unanimously agree that the proposed information-theoretic approach to interpret diffusion models is a novel and promising direction. We are further glad that reviewers found our experiments to be extensive (Tpva, hzC5), consider this work impactful for identifying the shortcomings of diffusion models (vbX2, Tpva, hzC5, THqw), and deemed our paper to be well-written (vbX2, Tpva, hzC5).

Some reviewers (vbX2, Tpva) pointed out a shortage of sufficiently complex examples that explore the application of PID in more challenging scenarios and to study relationships between more diverse lexical entities. We agree, and address this by providing three sets of visuals in the attached PDF:
1. **Figs 1, 2**: Visuals for more complex prompts taken from [1] similar to those in the diffusion training distribution. These prompts usually mention several objects and their corresponding attributes. We find that our method remains effective and informative even in these challenging examples. We visualize the information maps between objects and attribute-defining terms as per Tpva's suggestion. In both cases, we observe a high synergy because the attribute modifies the object's visual properties in some form.

2. **Figs 1, 2, 3, 4**: Visuals for CPID (Conditional PID). In this case, we extend the concept of PID for the conditional case where all the PID components and probability terms are now conditioned on the rest of the prompt. This is similar to the CMI extension of MI in [2]. We rewrite the equations from the main text with the required changes below for easy reference (we follow the same notations and definitions with an additional variable of $y$ signifying the rest of the prompt with the terms $y_1$ and $y_2$ removed):
    \begin{align}
    i(y_1, y_2; x | y) &= r(y_1, y_2; x | y) +  u(y_1 \backslash y_2; x | y) + u(y_2 \backslash y_1; x | y) + s(y_1, y_2; x | y) \\
\end{align}
 \begin{align}
    r(y_1, y_2; x | y) &= \min_{y_i} [-log\ p(y_i | y)] - \min_{y_i} [-log\ p(x|y_i, y) + log\ p(x | y) - log\ p(y_i | y)] \\
\end{align}
 \begin{align}
    u(y_1 \backslash y_2; x | y) &= i(y_1; x | y) - r(y_1, y_2; x | y) \\
\end{align}
 \begin{align}
    s(y_1, y_2; x | y) &= i(y_1, y_2; x | y) - r(y_1, y_2; x | y) -  u(y_1 \backslash y_2; x | y) - u(y_2 \backslash y_1; x | y)
    \end{align}

    We found that CPID provides slightly better localized results than its PID counterpart in practice as can be seen in **Figs 1 and 2**. This is expected as CPID accounts for the contribution of the rest of the prompt as the context in the image generation process and better captures the specific contribution of the two terms under consideration.

Some reviewers (hzC5, THqw) also requested further clarification of Eq 2 from the main text. To address THqw's query, we provide the derivation of the simplification done based on the orthogonality principle in the response to THqw.

To further expand on the estimator's uncertainty as mentioned by hzC5:

1. There is no guarantee that the estimator provides an upper or lower bound for the PID terms. It is dependent on the conditional and unconditional denoising MMSEs obtained from the diffusion model which is assumed to be an optimal denoiser for our experiments. However, in practice, this assumption need not hold because a neural network trained on a regression problem to minimize MSE need not converge to the global minima and instead may converge to a local minima. That being said, neural networks have been found to do really well on regression problems and the diffusion model, specifically, has been found to perform well on the denoising problem. Thus, we expect reasonable estimates despite the inherent uncertainty.

2. A measure of uncertainty is also introduced based on the number of samples under the same noise level, $\alpha$, in Eq 2's expectation term and from the number of $\alpha$ values sampled to evaluate the integral in the equation. Thus, we can obtain more confident information maps by using higher values for both of these hyperparameters. We provide visuals of the information maps for varying values of these hyperparameters on the "cat and elephant" sample from the COCO co-hyponym experiment (Fig 5 in the main text) in **Fig 7** in the attached PDF. We observe that the maps depict the same information, i.e., they are highly activated in the same regions, across variations but do become less noisy at higher values.

We provide a figure, **Fig 5**, in the attached PDF to complement our explanation of PID as defined in Eq 5 as per hzC5's suggestion

[1] Stablesemantics: A synthetic language-vision dataset of semantic representations in naturalistic images. 2024.

[2] Xianghao Kong, Ollie Liu, Han Li, Dani Yogatama, and Greg Ver Steeg. Interpretable diffusion via information decomposition. arXiv preprint arXiv:2310.07972, 2023.

---

### Decision · Program_Chairs · 2024-09-25

**Decision:**

Accept (poster)

**Comment:**

This paper introduces Partial Information Decomposition (PID) to interpret text-to-image diffusion models by decomposing input text prompts into their elementary components. The approach focuses on analyzing how individual tokens and their interactions through redundancy, uniqueness, and synergy influence the visual output of diffusion models. This method aims to provide insights into the model's handling of complex visual-semantic relationships and potential biases.

According to reviewers' comments, it is the first to apply PID in this context, offering a new lens to understand diffusion models. Its strengths include clarity, originality, and comprehensive experiments that illustrate its effectiveness in addressing biases and enhancing model interpretability. Reviewers praised the extensive visualizations and the practical application of the method in identifying and mitigating biases.

Reviewers pointed out several areas needing improvement: the paper could extend its analysis beyond simple text conditions and consider more complex interactions. There is also a call for clarity in some figures and equations, and concerns about the practical applicability of the results. Addressing these issues, particularly enhancing the robustness of the experimental setup and refining the clarity of visual representations, could significantly strengthen the paper.